## RESEARCH ARTICLE

# Mitotic reactivation and transcriptional bursting govern transcriptional noise in the early *Drosophila* embryo

Louise Maillard[1], Virginia L. Pimmett[1,*], Maria Douaihy[1,2,*], Pablo Garcia-Idieder[1], Rachel Topno[2], Amelie Brun[1], Antonio Trullo[1], Ovidiu Radulescu[2,‡] and Mounia Lagha[1,‡]

## ABSTRACT

Isogenic cells often exhibit variability in gene expression. Because of its stochastic nature, transcription is thought to lie at the heart of gene expression variability. While beneficial in some contexts, this transcriptional noise must be dampened to ensure the deployment of accurate gene expression programs. Here, we have investigated how the *cis*-regulatory code affects inter-nuclear variability in transcription, quantified at the level of individual RNA Polymerase II initiation events. Combining quantitative imaging in *Drosophila* embryos with mathematical modeling, we discovered that transcriptional noise is time dependent with two major components: mitotic exit and transcriptional bursting. Transcriptional noise peaks after mitosis due to the stochastic timing of postmitotic reactivation and the temporal heterogeneity of transcriptional bursting. When a steady-state regime is reached, transcriptional noise is then primarily driven by the kinetics of bursting. We demonstrate that the TATA box is necessary and sufficient to generate transcriptional noise, and assess the contribution of shadow enhancers to noise. Our work reveals the major contribution of the first polymerase passage after mitotic exit to transcriptional noise.

KEY WORDS: Transcriptional noise, Enhancer, Promoter, Development, *Drosophila*

## INTRODUCTION

Single-cell measurement techniques have revolutionized our view of transcription. In all organisms examined so far, transcription appears stochastic, marked by active periods (bursts of mRNA production) interspersed by inactive periods of various timescales (Lammers et al., 2020b).

In multicellular organisms, stochastic transcription can generate significant differences in mRNA levels between neighboring cells of an isogenic population. This type of gene expression variability is referred to as transcriptional noise. When amplified, transcriptional noise has the potential to significantly alter cell fate decisions. In the

[1]Institut de Génétique Moléculaire de Montpellier, Université Montpellier, CNRS, Montpellier 34293, France. [2]Laboratory of Pathogens and Host Immunity, Université Montpellier, CNRS, INSERM, Montpellier 34095, France.
*These authors contributed equally to this work

‡Authors for correspondence (mounia.lagha@igmm.cnrs.fr; ovidiu.radulescu@umontpellier.fr)

L.M., 0009-0009-9346-1976; V.L.P., 0000-0002-5786-7989; M.D., 0009-0002-7258-7413; P.G.-I., 0000-0001-7248-728X; R.T., 0000-0002-4920-9599; A.B., 0009-0009-2974-4445; A.T., 0000-0001-7560-3859; O.R., 0000-0001-6453-5707; M.L., 0000-0002-7082-1950

context of stochastic cell fate decisions, transcriptional noise is exploited to generate cellular plasticity and adaptation (Voortman and Johnston, 2022). For example, the stochastic selection of a single allele among the 1400 olfactory receptor genes in the mouse olfactory system enables remarkable receptor diversity (Ordway et al., 2024; Yusuf and Monahan, 2024). While beneficial in some contexts, transcriptional noise must generally be buffered during development to ensure the deployment of reproducible cell fate specification programs. Under normal conditions, developmental genes display a wide range of expression variability. In the fast-developing syncytial *Drosophila* embryo, developmental genes generally exhibit low transcriptional noise, maintaining consistent mature mRNA levels within a tissue (Boettiger and Levine, 2013; Lagha et al., 2013; Little et al., 2013). However, this homogeneity is not always reached during initial gene activation but can be shaped progressively throughout embryogenesis, as exemplified by the canalization of gap gene expression in *Drosophila* (Manu et al., 2009a,b). In zebrafish embryos, developmental genes are initially activated with a high level of transcriptional noise that progressively diminishes through cell cycle lengthening (Stapel et al., 2017). Nonetheless, under stress or in sensitized genetic backgrounds, gene expression becomes far more variable, often leading to incomplete penetrance of mutant phenotypes (Raj et al., 2010).

Transcriptional bursting is considered a major source of transcriptional noise (Zoller et al., 2015). Broadly, the modulation of transcriptional noise, whether amplified or attenuated, depends on the number and timescale of the rate-limiting steps involved in transcription. Regardless of specific architectures designed to filter or amplify noise, a few slow processes typically enhance noise, while numerous fast processes tend to diminish it (Radulescu et al., 2012). However, specific transcriptional regulatory mechanisms can enhance noise or, conversely, buffer the inherent variability in gene expression (Pal and Dhar, 2024). Here, we have investigated how the *cis*-regulatory code affects transcriptional noise, specifically focusing on the impact of shadow enhancers and promoter architecture (Fig. 1A).

In multicellular organisms, the expression of developmental genes is regulated by one or more enhancers with activities that may partially or fully overlap in space or time (Hong et al., 2008). Such enhancers are referred to as shadow enhancers (Kvon et al., 2021). Many examples from *Drosophila* to mammals have shown that shadow enhancers confer precision in gene expression and ensure phenotypic robustness (Frankel et al., 2010; Lagha et al., 2012; Perry et al., 2010). We therefore sought to ask how they affect transcriptional noise. A previous study showed that a pair of shadow enhancers can better buffer the fluctuations in input regulatory factors than a simple duplication of a single enhancer (Waymack et al., 2020). However, this study did not examine the impact of shadow enhancers on temporal noise, nor did it investigate the effect of distally located enhancers. The contribution of shadow enhancers to transcriptional noise thus remains an unanswered question. In addition to transcription factor decoding by enhancers, the resulting

*DEVELOPMENT*

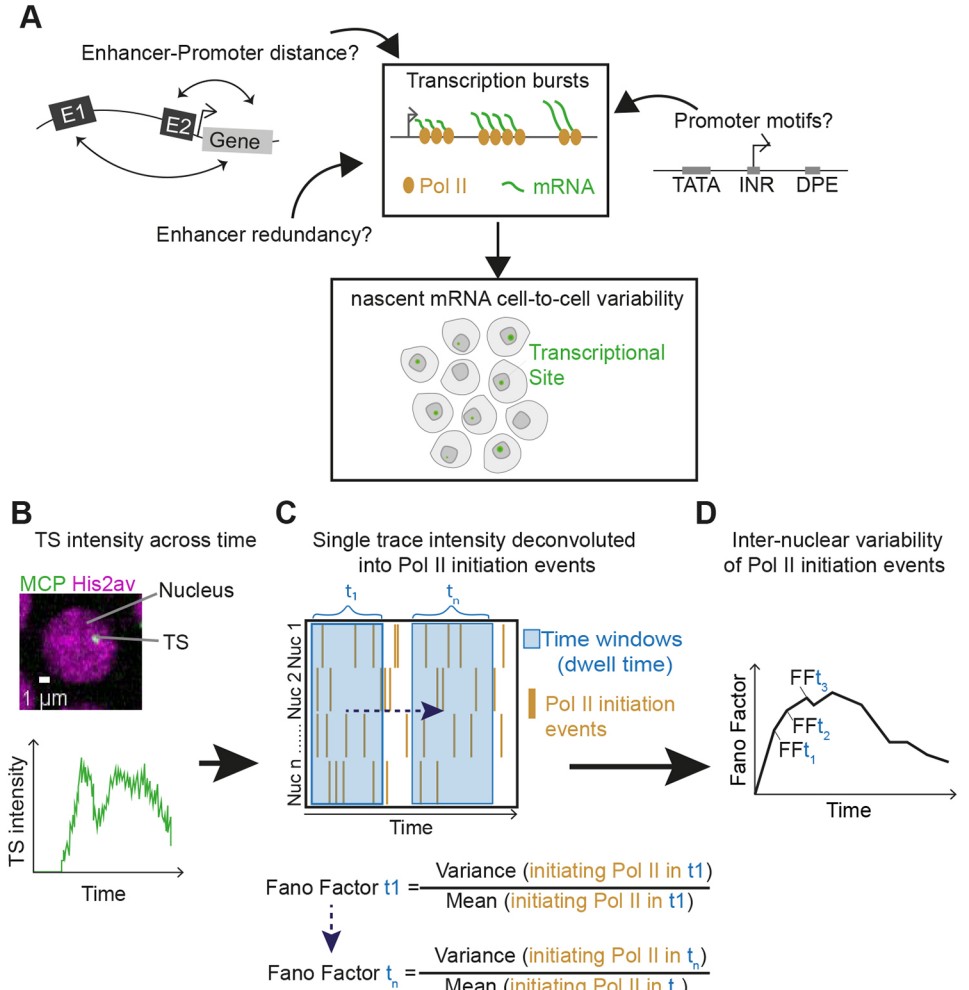

**Fig. 1. Quantifying the instantaneous inter-nuclear variability in transcription.** (A) Schematic illustrating potential sources of nascent mRNA inter-nuclear variability (transcriptional noise). (B-D) Analysis pipeline to quantify instantaneous inter-nuclear variability within a tissue. (B) Transcriptional intensity of each transcription site (TS, green) over time (below) per nucleus (magenta). (C) Heatmap showing the number of Pol II initiation events for each nucleus across time after deconvolution. Inter-nuclear variability is calculated across a moving time window in nc14 from pooled nuclei in a single embryo using the Fano factor (FF) of Pol II initiation events during nc14. The moving time window has a width equal to the Pol II dwell time (typically 3-6 min) centered at intervals of 1 min. (D) Inter-nuclear variability in transcriptional activity is plotted as the mean FF across biological replicates (different embryos) for each window.

transcriptional outcome highly depends on the promoter sequence. The contribution of promoter motifs to transcriptional noise has been well studied (Blake et al., 2006; Larsson et al., 2019; Ramalingam et al., 2021), although mainly through fixed approaches that cannot fully capture how transcriptional dynamics influence noise.

In this work, we employed live imaging of transcription and computational approaches to ask how shadow enhancers and promoter motifs affect transcriptional noise. We focused on three distinct model loci and characterized the inter-nuclear variability in Pol II initiation events during *Drosophila* dorso-ventral patterning. We found that the transcriptional noise is time dependent and occurs via two distinct phases. A first phase begins shortly after mitotic exit, during which noise peaks and is influenced by the stochasticity of postmitotic reactivation timings. A second phase occurs later in the cell cycle when noise reaches a steady state and is primarily determined by the transcriptional bursting parameters. At steady state, noise varies between shadow enhancers, and is increased by the TATA motif at the promoter but remains unaffected by INR and DPE motifs. In support of these findings, we provide a mathematical model relating noise to bursting and postmitotic reactivation parameters.

## RESULTS
### A quantitative approach to estimate inter-nuclear variability in Pol II transcription initiation events
Live imaging of transcription can provide access to the evolution of single allele transcriptional activity across time. We reasoned

that we could exploit this property to quantify inter-nuclear variability in nascent mRNA production across time. We used the MS2/MCP mRNA labeling system and imaged transcription within the first 30 min of nuclear cycle 14 (nc14) in *Drosophila* embryos (Fig. 1B). With such movies and after calibrating our signal, we employed BurstDECONV (Douaihy et al., 2023), a method able to infer individual polymerase initiation events for each nucleus. To quantify the temporal variability in transcriptional activity, now measured with absolute counts of RNA polymerase II (Pol II) initiation events, we binned our movies into sliding time windows. For each window, we computed the Fano factor (FF), defined as the variance divided by the mean of the number of Pol II initiation events within the window (see Materials and Methods, and Fig. 1C). This allows us to obtain the instantaneous inter-nuclear variability of Pol II initiation events over time (Fig. 1D).

The Fano factor serves as a measure of bursting, as it equals one in the absence of bursting (i.e. under Poisson noise) and increases with the amount of burstiness (see Materials and Methods). The use of FF rather than the coefficient of variation (CV) is appropriate, as it applies to the number of events in a given time interval and allows testing for deviations from a Poisson distribution. The Fano factor captures different sources of variability in RNA production. As will be shown later in this study, there are two main sources: transcriptional bursting and the stochastic postmitotic reactivation of transcription.

## Enhancers with overlapping functions differentially affect transcriptional noise

To investigate how enhancers with overlapping activities affect transcriptional noise, we focused on two model genes, *short gastrulation* (*sog*) and *snail* (*sna*), both known to be co-regulated by separate proximal and distal enhancers. Both gene products are activated by the morphogen Dorsal and are essential for patterning the dorso-ventral axis (Leptin, 1991). *Sog* encodes a secreted inhibitor crucial for establishing the Decapentaplegic (Dpp) morphogen gradient (analogous to Chordin and BMP in vertebrates, respectively). In the early *Drosophila* embryo, *sog* contributes to specifying the presumptive neurogenic and dorsal ectoderm fates. *sna* encodes a conserved transcription factor that regulates mesodermal fate (Leptin, 1991).

We employed a series of *sog*-MS2 CRISPR alleles (Whitney et al., 2022) to examine how endogenous enhancers affect noise. Specifically, three alleles were used: one containing both enhancers (*sog^{WT}*), one containing a deletion of the proximal intronic enhancer (*sog^{ΔProx}*) and one containing a deletion of the distal enhancer located 20 kb upstream of the *sog* transcription start site TSS (*sog^{ΔDist}*; Fig. 2A). These alleles were previously characterized as functionally partially redundant, although they differ slightly in spatiotemporal expression pattern, strength and phenotypes (Whitney et al., 2022). However, inter-nuclear variability in gene expression was not addressed. In order to precisely quantify temporal noise, we imaged these three alleles with high temporal resolution and within a common spatial domain, the ventro-lateral region encompassing the presumptive neurectoderm (Fig. 2B). In this domain during nc14, transcriptional sites show a significant difference in behavior, with *sog^{ΔProx}* demonstrating higher expression than *sog^{WT}* or *sog^{ΔDist}* (Fig. 2C; Fig. S1A,B). In addition, nuclei expressing *sog* from the distal enhancer only (*sog^{ΔProx}*) produce more variable transcription than those driven by both enhancers (*sog^{WT}*) or the proximal enhancer alone (*sog^{ΔDist}*) (see *y*-axis Fig. 2D). To quantify this variability, we used BurstDECONV and produced Pol II placement maps for each genotype (Fig. 2E). Temporal noise, as scored by the FF, is much higher for *sog^{ΔProx}* nuclei. For example, at 20 min in nc14, the *sog^{ΔProx}* nuclei show a FF that is four-fold higher than that of an equivalent wild-type population (Fig. 2E). In contrast, nuclei driven only from the proximal enhancer and lacking the distal enhancer (*sog^{ΔDist}*) are much less variable.

In addition to the quantitative comparison at a given time point, our approach also provides information on how this inter-nuclear variability evolves across time. For both *sog^{WT}* and *sog^{ΔDist}* nuclei, the FF remains relatively stable within nc14. In contrast, the FF of the *sog^{ΔProx}* population exhibits a continuous increase until it peaks at around 20 min, then slowly decreases to reach a plateau (Fig. 2E). Thus, the *sog^{ΔProx}* displays more variability at two levels: the absolute FF at any given point but also in its evolution over time.

Since the overall transcription of the *sog^{ΔProx}* allele is stronger than the two other genotypes (Fig. 2C, Fig. S1A-C), we next asked if the noise profiles were simply paralleling the underlying transcriptional strength. To test this hypothesis, we examined transcription dynamics from the *sna* locus. Like *sog*, *sna* is also regulated by two enhancers: a proximal enhancer and a distal (shadow) enhancer located 8 kb away. We used existing *sna BAC-MS2* reporter constructs (Bothma et al., 2015) (Fig. 2F) containing 25 kb of the *sna* locus with either both enhancers intact (*sna^{WT}*) or with one of the two shadow enhancers replaced by an alternative sequence, denoted here as *sna^{ΔDist}* (i.e. driven exclusively by the proximal enhancer) and *sna^{ΔProx}* (i.e. driven exclusively by the distal enhancer). The analysis was performed in the central part of the mesoderm at peak levels of input Dorsal concentrations in order to minimize the effect of spatial regulation (Fig. 2G). In contrast to the *sog* allelic series, there is no qualitative difference in single nuclei traces between the *sna^{WT}* reporter and *sna^{ΔProx}* (Fig. 2I). The average transcriptional intensity profiles show similar behaviors, with an initially increasing expression followed by an attenuation that plateaus after 20 min into nc14 (Fig. 2H). Quantitatively, average intensities as well as timing of transcriptional activations appear indistinguishable between the wild-type control (*sna^{WT}*) and *sna^{ΔProx}* (Fig. 2H,I; Fig. S1D-F). In contrast, when driven by the proximal enhancer only (*sna^{ΔDist}*), *sna* transcription is reduced (Fig. 2H; Fig. S1D-F).

Next, we calculated the FF across time (Fig. 2J). The *sna^{WT}* and *sna^{ΔProx}* genotypes exhibit an overall similar noise profile, with an initial peak shortly after mitosis followed by a steady state reduced noise behavior (Fig. 2J). In contrast, *sna^{ΔDist}* does not show this initial boost of noise but stays relatively constant at a slightly elevated FF. Note that FF and mean expression levels are not always correlated, as evidenced with the *sna-MS2* data but also for other genotypes tested (Fig. S5). We therefore conclude that transcriptional noise, examined in terms of Pol II initiation events, is decoupled from the overall enhancer strength.

In addition, comparing these two loci allowed us to investigate how enhancer distance could impact inter-nuclear variability in gene expression. In principle, enhancer positioning could affect the probability of enhancer-promoter encounters with potential consequences on transcription noise. From our reduced number of model loci, we do not observe higher noise when enhancer-promoter distances are increased. Moreover, from our two model loci and four enhancers, it seems that the presence of multiple co-acting enhancers does not obviously contribute to noise reduction. However, given the small number of enhancers examined here, we cannot draw general conclusions on the effect of shadow enhancers on transcriptional noise.

## Postmitotic reactivation timing represents a major source of transcriptional variability

While examining how inter-nuclear variability in transcriptional activity varies across time, we noticed a recurrent peak of noise occurring during the first 10 min of nc14 (Fig. 2E,J). This early phase of noise is not peculiar to nc14 as it is also detected in nc13 (Fig. S1G). Although early mitotic divisions of *Drosophila* embryos are synchronized, not all nuclei activate transcription simultaneously upon exiting mitosis. We therefore sought to investigate how this variable could affect transcriptional noise. The time to first activation, referred to as the mitotic lag time, is stochastic but highly dependent on the *cis*-regulatory code (Dufourt et al., 2018; Fernandes et al., 2022; Yamada et al., 2019). This timing has been shown to contribute to the temporal coordination in gene expression (synchrony) (Dufourt et al., 2018; Yamada et al., 2019) and could depend on mitotic bookmarking (Bellec et al., 2022; Ferraro et al., 2016; Hsiung et al., 2016). However, how reactivation times after mitosis contribute to noise has thus far remained unknown. In order to examine the impact of mitotic lag time on noise, we artificially synchronized activation by temporally aligning all traces to the first detected activation and plotted the temporal FF of Pol II initiation events (see Materials and Methods). By doing so, the initial spike of noise disappears and FF is constant across time, yet continues to differ between genotypes (Fig. 3A-C,F-H). This is particularly evident for *sog^{ΔProx}*, where the peak FF (at 18 min) is no longer detectable after removing the mitotic lag time component (Fig. 3G,H). After this temporal realignment, the inter-nuclear noise for this genotype remains constant throughout the cycle (Fig. 3H).

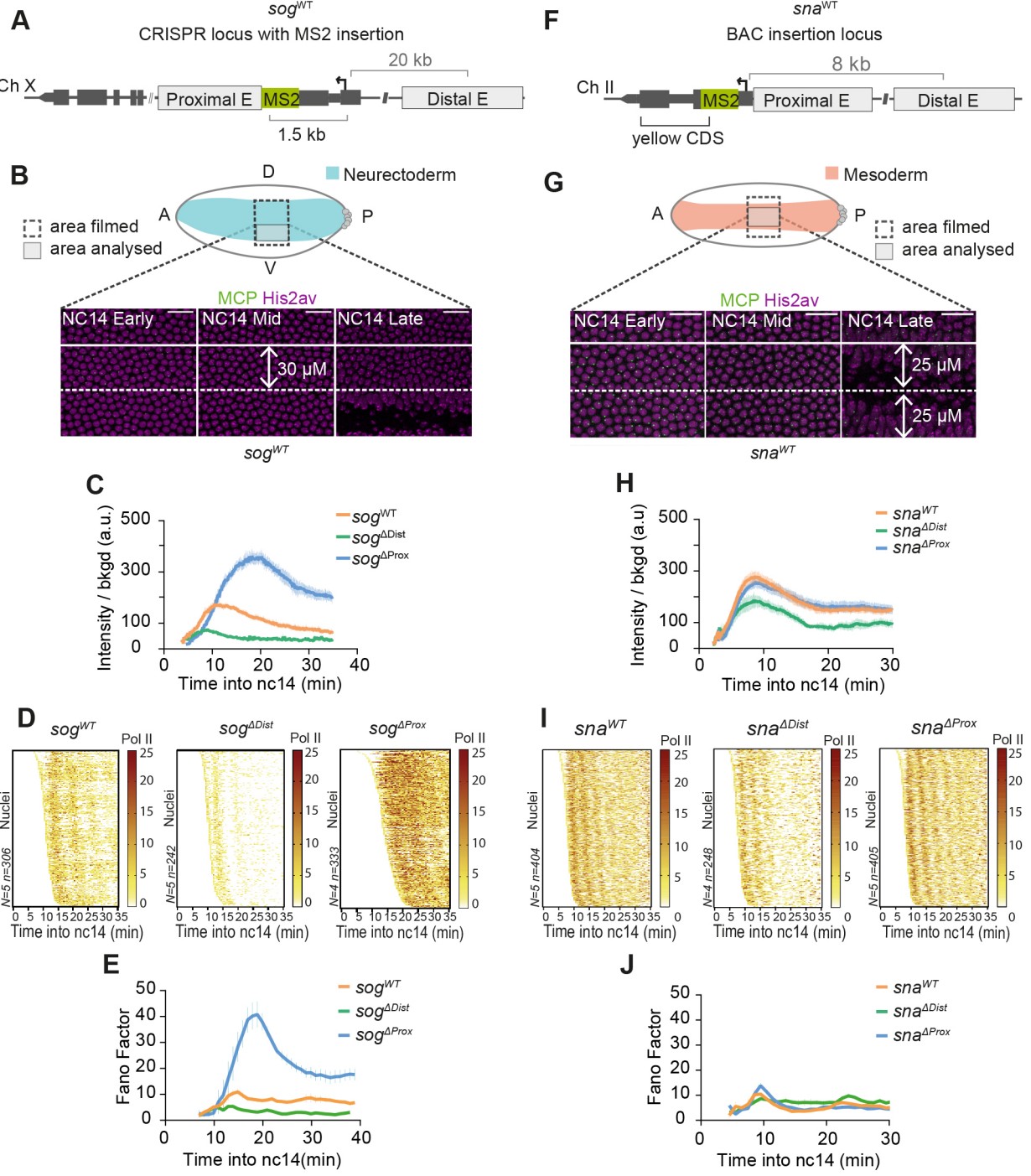

**Fig. 2. Redundant enhancers differentially affect transcriptional noise.** (A,F) Schematics of *sog^WT^* and *sna^WT^* MS2 alleles. (B,G) Spatial nuclear selection at indicated time points. Scale bars: 10 µm. (B) Nuclei analyzed within 30 µm of the mesoderm-neurectoderm boundary for *sog*. (G) Nuclei analyzed within ±25 µm of the gastrulation furrow for *sna*. (C,H) Average active transcription site intensity across time by genotype (mean±s.e.m.). (D,I) Heatmaps representing the number of polymerase initiation events across nc14. Each row corresponds to an individual nucleus with embryos pooled. (E,J) Fano factor of Pol II initiation events during nc14 (mean±s.e.m.) by genotype calculated across the moving time window. *sog^WT^* N=5 embryos, *n*=306 nuclei; *sog^ΔDist^* N=5 embryos, *n*=242 nuclei; *sog^ΔProx^* N=4 embryos, *n*=333 nuclei; *sna^WT^* N=5 embryos, *n*=404 nuclei; *sna^ΔDist^* N=4 embryos, *n*=248 nuclei; *sna^ΔProx^* N=5 embryos, *n*=405 nuclei. See also Movies 1-6.

These results suggest that the mitotic lag time represents a major source of inter-nuclear variability in transcriptional activity. The contribution of various sources is variable in time. At short times after mitosis, the post-mitotic lag time and time inhomogeneity of the bursting kinetics can contribute to 50-75% of the peak FF. At steady state, the only source of noise is bursting kinetics (see Fig. 3B,C,G,H).

In the past, we and others have modeled the distribution of the mitotic lag time (Alamos et al., 2023; Bellec et al., 2022; Dufourt et al., 2018; Lammers et al., 2020a). In a simple 'staircase' model, upon mitotic exit each nucleus is considered to irreversibly 'travel' through a number of rate-limiting steps before reaching a competent ON state from which it activates transcription for the first time in the cell cycle.

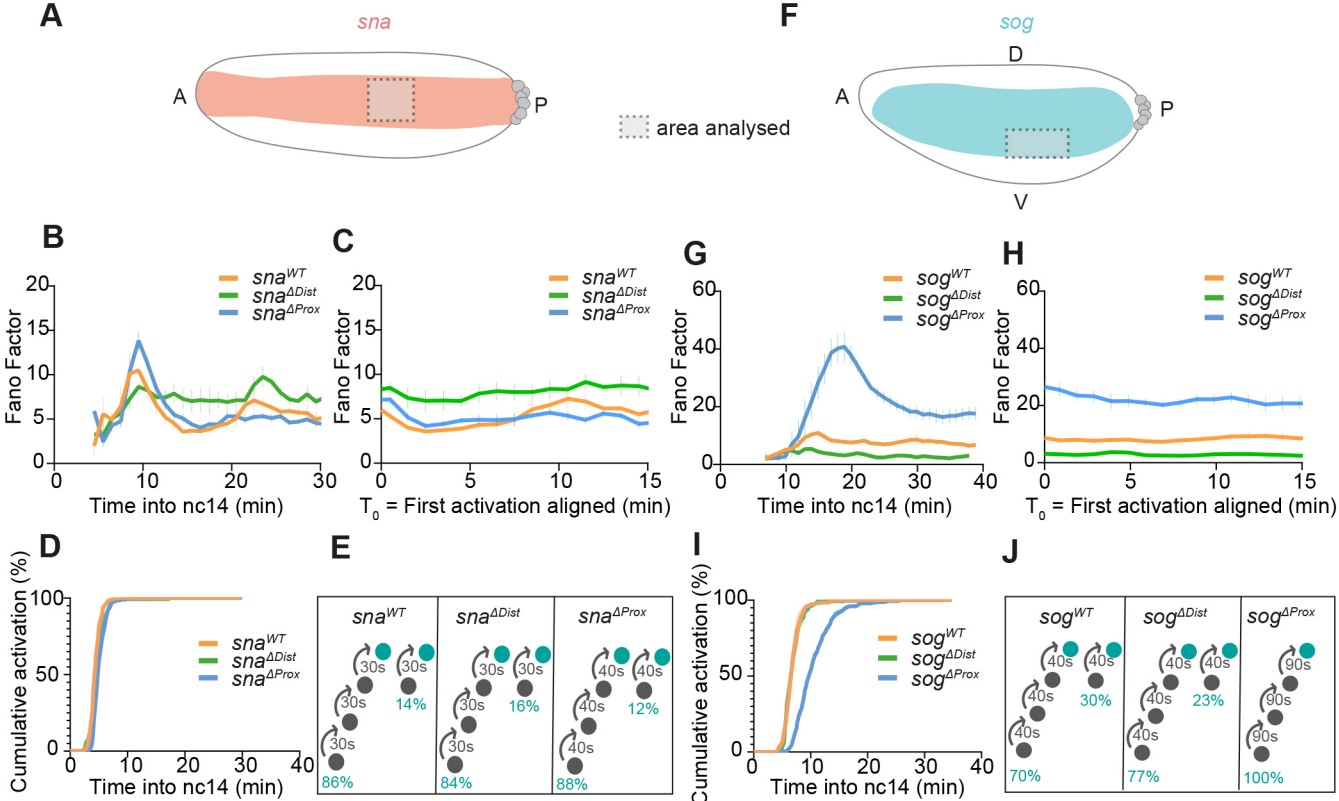

**Fig. 3. Non-synchronous reactivation is an important source of inter-nuclear transcriptional variability.** (A,F) Schematics indicating the genotype and the spatial domain of interest. (B,C,G,H) Fano factor of Pol II initiation events (mean±s.e.m.) across nc14: (B,G) aligned at mitosis ($T_0$=mitosis) or (C,H) aligned at first detected transcription of each nucleus. (D,I) Cumulative activation percentage (mean±s.e.m.) by genotype. (E,J) Schematic showing the number of steps required to reach activation after mitosis for indicated genotypes. Note that B and G are identical to Fig. 2E,J. $sog^{WT}$ N=5 embryos, n=306 nuclei; $sog^{\Delta Dist}$ N=5 embryos, n=242 nuclei; $sog^{\Delta Prox}$ N=4 embryos, n=333 nuclei; $sna^{WT}$ N=5 embryos, n=404 nuclei; $sna^{\Delta Dist}$ N=4 embryos, n=248 nuclei; $sna^{\Delta Prox}$ N=5 embryos, n=405 nuclei. See also Movies 1-6.

To gain insight into the quantitative differences in this mitotic lag following enhancer deletion, we modeled the distributions of times to first activation. We utilized an N-component mixed Erlang distribution that copes with the fact that nuclei pass through several limiting steps prior to activation, with each step having the same duration.

We first analyzed the *sna* group, as the genotypes showed no observable differences in activation timing post-mitosis (Fig. 3D). For all *sna* genotypes, regardless of the presence of one or two enhancers, the mitotic lag time distribution can be modeled with similar trajectories. The majority of nuclei travel through three short-lived rate-limiting states prior to activation (>80% of nuclei), while a subset of the population (12-16%) reaches the competent active state via a single transition (Fig. 3E). These results suggest that mitotic lags are not obviously affected by the deletion of the *sna* proximal or distal enhancer when assessed in the context of a reporter BAC transgene.

Contrary to *sna*, enhancer deletion significantly increases the mitotic lag for the *sog* group (see $sog^{\Delta Prox}$, Fig. 3I). Thus, we modeled the mitotic lag time distribution for the different *sog* alleles. In the presumptive neurectoderm, when both enhancers are present ($sog^{WT}$), our modeling suggests that nuclei traverse two possible paths (Fig. 3J). In the first scenario, the majority of nuclei (70%) transit through three distinct states of 40 s each. A second scenario applies to 30% of the $sog^{WT}$ population that requires only one unique transition of 40 s before reaching

activation. Interestingly, upon deletion of the distal enhancer, the distribution of the time to first activation is similar to wild type, and modeling shows parameters resembling the $sog^{WT}$ population. Contrary to these two genotypes, $sog^{\Delta Prox}$ nuclei show a more homogeneous distribution, with only one path that employs much longer steps (90 s instead of 40 s) (Fig. 3J). We hypothesize that these longer transitions prior to activation are responsible at least in part for the high level of transcriptional noise during the first 10 min of nc14. Taken together, our results show that the mitotic lag represents an important source of noise in transcriptional activity.

## Spatial differences in transcriptional noise

We next asked how spatial regulatory input could affect inter-nuclear variation in transcriptional activity. To compare spatial differences in transcriptional noise, we divided the *sog* pattern into four distinct zones based on distance from the mesoderm-neurectoderm border at mid-nc14 (Whitney et al., 2022). These domains are the ventral-most region (the presumptive mesoderm), where *sog* becomes transcriptionally silenced during nc14, the ventro-lateral region, the dorso-lateral region and the dorsal region (Fig. 4A).

Given that transcriptional signals are calibrated, we can directly compare the temporal evolution in FF for the same genotype in different spatial domains. When both enhancers are present, *sog* transcription within the lateral domain is homogeneous. Indeed, temporal FF curves in dorso-lateral and ventro-lateral domains are overlapping (Fig. S2A,B), even when the mitotic lag time is

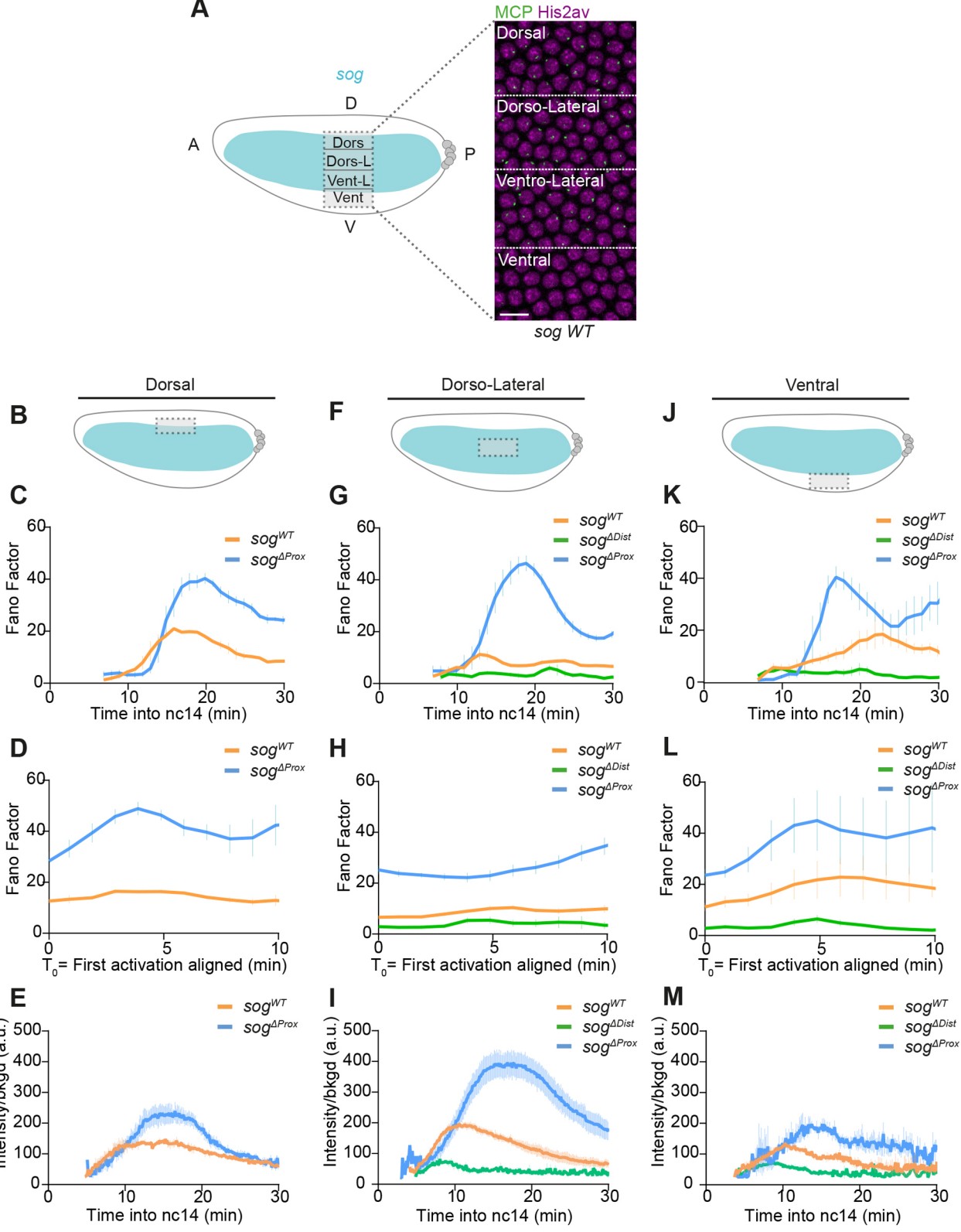

**Fig. 4. Quantifying transcriptional noise across space.** (A) Schematic showing *sog* expression regions at 20 min in nc14. Regions were defined at 30 μm intervals from the mesoderm-neurectoderm boundary. Scale bar: 10 μm. (B,F,J) Schematics representing the area of interest in the panels below. (C,D,G,H, K,L) Fano factor of Pol II initiation events during nc14 (mean±s.e.m.) calculated across the moving time window: (C,G,K) after mitosis ($T_0$=mitosis) or (D,H,L) after aligning the first activation of each nucleus to time $T_0$. (E,I,M) Average active transcriptional intensity (mean±s.e.m.) for indicated domains and genotypes. Dorsal region: $sog^{WT}$ N=2 embryos, n=126 nuclei; $sog^{\Delta Prox}$ N=4 embryos, n=165 nuclei. Dorso-lateral region: $sog^{WT}$ N=4 embryos, n=286 nuclei; $sog^{\Delta Dist}$ N=3 embryos, n=84 nuclei; $sog^{\Delta Prox}$ N=5 embryos, n=407 nuclei. Ventro-lateral region: $sog^{WT}$ N=5 embryos, n=306 nuclei; $sog^{\Delta Dist}$ N=5 embryos, n=242 nuclei; $sog^{\Delta Prox}$ N=4 embryos, n=333 nuclei. Ventral region: $sog^{WT}$ N=3 embryos, n=140 nuclei; $sog^{\Delta Dist}$ N=5 embryos, n=220 nuclei; $sog^{\Delta Prox}$ N=3 embryos, n=121 nuclei. See also Movies 1-3.

removed by aligning traces at first activation (Fig. S2C). Instead, when examined in the ventral or the dorsal domain, inter-nuclear variability in *sog* transcription is much more prominent (Fig. S2B,C). In both of these regions, *sog* is known to be regulated by repressors that silence *sog* transcription (Forbes Beadle et al., 2023; Koromila and Stathopoulos, 2019). In the ventral region, the repressor Sna directly represses *sog* expression (Bothma et al., 2011; Esposito et al., 2016). Given the timing at which *sog* noise peaks in nc14 in the presumptive mesoderm, it is likely that it is associated with new regulation of *sog*, such as Sna-mediated repression.

We asked how transcriptional noise upon enhancer deletion would be affected in space. Consistent with what has been previously reported (Whitney et al., 2022), in all domains, nuclei expressing *sog* from the distal enhancer alone (*sog^{ΔProx}*) tend to exhibit a higher level of transcription, but also tend to be activated later than *sog^{WT}* nuclei (Fig. S2A-C). After deconvolving single nuclei traces and calculating the FF, nuclei transcribing from the distal *sog* enhancer (*sog^{ΔProx}*) show a high level of noise in the presumptive mesoderm, the dorso-lateral region and the dorsal region, similar to what was observed in the ventro-lateral domain (Fig. 4B,C,F,G,J,K). When temporally aligning traces at first activation, the noise spike disappears but the *sog^{ΔProx}* allele continues to exhibit a significant level of inter-nuclear noise (Fig. 4D,H,L). We note that, in addition to this inter-nuclear variability, we observe an important inter-embryonic variability when imaging the *sog^{ΔProx}* allele in the ventral domain (presumptive mesoderm), as evidenced by the large error bars (Fig. 4E,I,M). This extra variability is probably due to the ongoing repression of the *sog* locus in the mesoderm, and the fact that the proximal and distal enhancer integrate this repressor input differentially (Dunipace et al., 2019; Whitney et al., 2022).

In conclusion, by using the *sog* locus as a model, we investigated how a morphogen, Dorsal, affects transcriptional noise in space and time. Regardless of Dorsal concentrations, postmitotic reactivation timing appears as an important source of transcriptional noise. In all spatial domains, the presence of two enhancers instead of a unique enhancer does not contribute to noise filtering.

### The TATA box motif generates transcriptional noise while the DPE and INR motifs only moderately impact it

While enhancers dictate where and when genes are turned on, their action is ultimately integrated at the promoter, which contains conserved promoter motifs such as the initiator (INR), the TATA box or the downstream promoter element (DPE) (Haberle and Stark, 2018; Sloutskin et al., 2021). Several studies have now established that these promoter motifs contribute differentially to coordinating the timing of transcriptional activation (Boettiger and Levine, 2013; Falo-Sanjuan et al., 2019; Lagha et al., 2013; Ling et al., 2019) and to bursting kinetics (Hoppe et al., 2020; Pimmett et al., 2021; Yokoshi et al., 2022). We therefore investigated how core promoter sequences affect transcriptional noise during early *Drosophila* development.

We used a previously established synthetic transgenic platform, containing various configurations of core promoters (100 bp) while sharing a common enhancer and a MS2 reporter (Pimmett et al., 2021) (Fig. 5A, Table S1). We first investigated the effect of the INR motif on inter-nuclear variability. The presence of an INR is often associated with promoter-proximal Pol II pausing (Hendrix et al., 2008; Shao et al., 2019). Modeling transcriptional dynamics from various model promoters suggested that pausing may not be uniformly imposed on each polymerase (Pimmett et al., 2021; Tantale et al., 2021). Given the stochasticity of pausing, we reasoned that this extra regulation step could affect inter-nuclear variability. We compared the *sna* core promoter to a mutant

containing an INR motif (*sna+INR*) and observed a moderate increase in temporal noise (Fig. 5B). In a more sensitized genetic background with reduced Dorsal levels in the embryo, the inter-nuclear variability does not obviously differ between genotypes (Fig. S3A-D). We also examined the inverse case using the naturally INR-containing *Kr* core promoter with two INR mutants (*Kr-INR1* and *Kr-INR2*), which also did not lead to an obvious effect on transcriptional noise (Fig. 5C). Taken together, these results suggest that the presence of an INR motif, and thus possible regulation by paused polymerase, do not significantly augment inter-nuclear variability in transcriptional activities.

Next, we investigated how the TATA box motif affects transcriptional noise. The TATA box in yeast and *Drosophila* has been shown to be associated with higher transcriptional variability (Blake et al., 2006; Larsson et al., 2019; Ramalingam et al., 2021; Raser and O'Shea, 2004). The *sna* core promoter contains a TATA box that is required to sustain high levels of transcription with mutations driving reduced reporter mRNA output across nc14 (Pimmett et al., 2021). The mutation of a TATA box resulted in lower inter-nuclear variability compared to the wild-type control (Fig. 5D), which we also confirmed with a parallel mutation in the TATA-containing *Kr* promoter (Fig. 5E).

To further explore these findings with a different enhancer/promoter pair, we created MS2 transgenes where transcription is driven by *twist* (*twi*) cis-regulatory sequences. To create a *twi* transcriptional reporter that closely recapitulated endogenous early embryonic *twi* expression, a sequence of 1.2 kb comprising two natural enhancers (distal and proximal) (Jiang et al., 1991) upstream of the *twi* core promoter was placed upstream of a 24xMS2 array and a *yellow* reporter gene (Fig. 5F). The wild-type *twi* promoter does not contain a TATA box but contains INR and DPE motifs. A previous study demonstrated that mutating the *twi* DPE motif (*twi-DPE*) strongly affects mRNA levels *in vitro*, but this effect could be partially rescued by adding a TATA box (*twi-DPE+TATA*) (Zehavi et al., 2014). These two *twi* promoter mutations were introduced in our MS2 reporter constructs (Table S2). In the context of early embryogenesis, we observed that mutating the *twi* DPE indeed strongly reduces transcription, but this effect is significantly rescued by adding a TATA box to the DPE-mutant promoter (Fig. S4A-C). Despite its strong effect on transcription capacity, the *twi* and *twi-DPE* mutant promoters exhibit similar levels of noise (Fig. 5G,H). In contrast, *twi-DPE+TATA* exhibits substantially higher noise, with an initial peak at 10 min, due to mitotic lag time (Fig. 5G,H).

To decipher how transcription bursting affects the inter-nuclear variability in transcription with the addition of a TATA box, we sought to extract promoter switching rates (Fig. 6A). Heatmaps of single Pol II initiation events are shown in Fig. 6B,C. Both genotypes show long-lived inactive periods (white intervals), particularly after 20 min into nc14. However, in the presence of a TATA box (*twi-DPE+TATA*), we also observe a higher density of Pol II events during the first 20 min. To gain a more quantitative view of transcription dynamics for these promoters, we examined the mean waiting time between successive Pol II initiation events. The mean transcription rates of the promoter evolve during nc14, progressively decreasing before reaching a plateau at around 20 min into nc14 (Fig. S4A). We therefore focused our attention on the 20-30 min period, during which transcription had reached steady state (Fig. S4A). For both genotypes, the distribution of waiting times between successive Pol II initiation events can be captured by a three-state model (Fig. S4D,E). We therefore considered a three-state promoter model comprising two inactive periods, OFF1 and OFF2, and an active ON period during which Pol II initiation occurs at a particular rate ($k_{ini}$) (Fig. 6D).

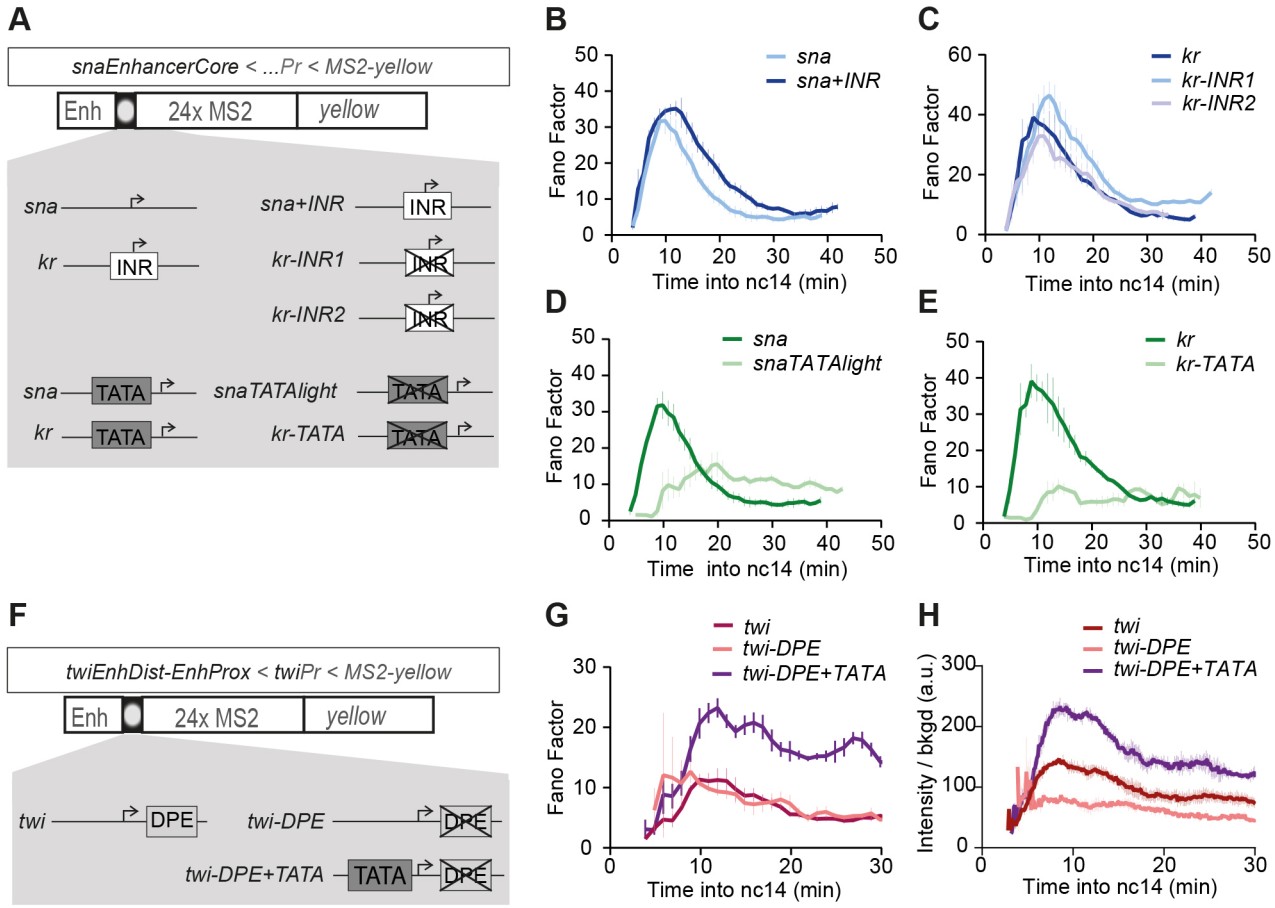

**Fig. 5. The TATA box promoter motif increases transcriptional noise while the INR and DPE motifs have only small contributions.** (A) Schematic of the promoter transgenes used to decipher the impact of the INR and TATA box motifs. For INR mutants, the *sna* core promoter contains the *kr* INR (*sna+INR*), and the *kr* INR motif was replaced by the *sna* TSS (*kr-INR1*) or *brk* TSS (*kr-INR2*). *snaTATAlight* contains a TATA box mutation to the non-canonical TATA box of *kr*; *kr-TATA* refers to a mutation of the first five bases of its non-canonical TATA box. (B-E) Fano factor of Pol II initiation events during nc14 (mean±s.e.m.) by genotype calculated across the moving time window, after mitosis ($T_0$=mitosis). (F) Constructs using *twi* E1/E2 with either the natural *twi* promoter, a mutated DPE (*twi-DPE*) and the addition of a TATA box motif (*twi-DPE+TATA*). (G) Fano factor of Pol II initiation events during nc14 (mean±s.e.m.) calculated across a moving time window ($T_0$=mitosis). (H) Average active transcriptional intensity (mean±s.e.m.) for indicated domains and genotypes. *sna* N=3 embryos, *n*=216 nuclei; *snaTATAlight* N=6 embryos, *n*=353 nuclei; *kr* N=4 embryos, *n*=243 nuclei; *kr-TATA* N=2 embryos, *n*=98 nuclei; *sna+INR* N=4 embryos, *n*=236 nuclei; *kr-INR1* N=5 embryos, *n*=342 nuclei; *kr-INR2* N=3 embryos, *n*=168 nuclei; *twi* N=3 embryos, *n*=485 nuclei; *twi-DPE* N=3 embryos, *n*=357 nuclei; *twi-DPE+TATA* N=3 embryos, *n*=451 nuclei. See also Movies 7-9 and Pimmett et al. (2021).

Although the data determine the number of states, they do not discriminate between sequential and non-sequential versions of three-state promoters that are entirely equivalent (Radulescu et al., 2025; Tantale et al., 2021). Our choice of a non-sequential model is therefore purely illustrative, and the alternative choice of a sequential model would not alter our conclusions.

The duration and probabilities are not significantly different when comparing the TATA containing promoter *twi-DPE+TATA* to the control *twi-DPE* (Fig. 6E,F, Table S3). However, we observe an important difference in initiation rates, with three-fold higher Pol II initiation rates in the presence of a TATA box (Fig. 6G, Table S3). We hypothesize that this difference in initiation rates may contribute to the increased transcriptional noise elicited by the TATA box. This hypothesis is supported by our analytical results relating kinetic parameters to the value of noise at steady state (see Materials and Methods, Table S3 and supplementary Materials and Methods). Based on these results, we conclude that, even in simple genomic contexts, where enhancers do not require looping and where input transcription factor regulators are at peak levels, promoter architecture remains crucial in modulating transcriptional noise.

## Mathematical modeling unravels the determinant causes of transcriptional noise

To gain deeper insights into the modulation of transcriptional noise, we developed several mathematical models for the sequence of transcription initiation events. In all these models, the first initiation occurs at a random time following mitotic exit and is governed by a mixed Erlang distribution, corresponding to the above-mentioned staircase model. Subsequent initiation events are positioned relative to this first event, assuming that the waiting times between successive events are independent, positive and randomly distributed according to the bursting parameters. When transcription occurs in bursts, the distribution of the waiting times separating successive initiation events is not exponential and the process is therefore non-Poissonian. In order to cover this general situation, we considered a renewal process, i.e. a stochastic process used to model events that occur over time where the times between consecutive events are independent and identically distributed (not necessarily Poissonian) random variables (see Materials and Methods). Using results from Radulescu et al. (2025), we were able to relate the first two moments of the distributions used to compute the Fano factor to the kinetic parameters of transcriptional

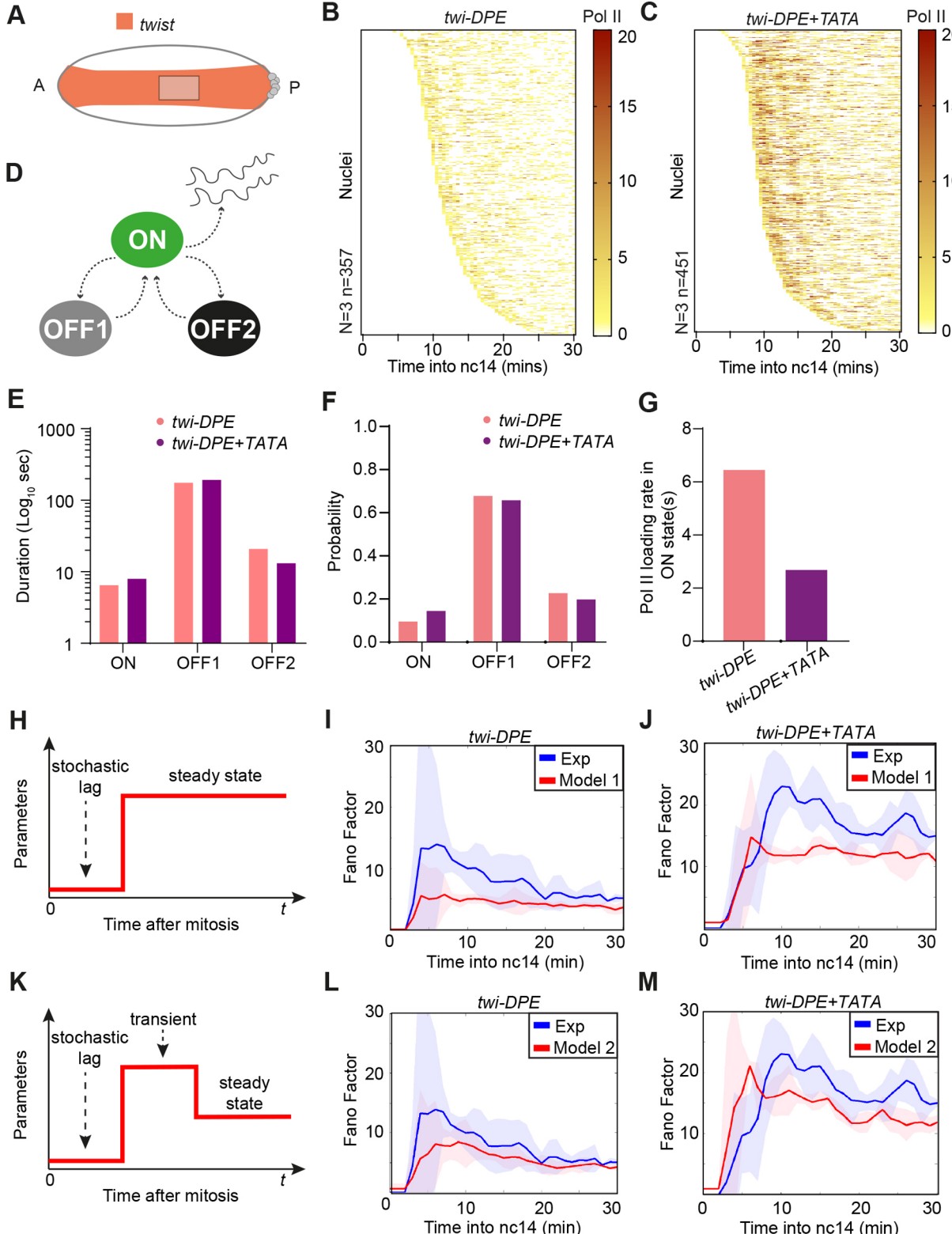

**Fig. 6. Mathematical modeling reveals the sources of transcriptional noise and its temporal inhomogeneity.** (A) Schematics representing the region of interest. (B,C) Heatmaps showing the number of polymerase initiation events, with each row representing a nucleus from pooled embryos. (D) Diagram representing a possible non-sequential three-states model used to extract bursting parameters in E-G. (E) Average duration of the ON, OFF1 and OFF2 states at steady state. (F) Probability to be in the ON, OFF1 and OFF2 states at steady state. (G) Average Pol II loading interval separating successive initiation events in the ON state at steady state. The loading interval is the inverse of the initiation rate $k_{ini}$. (H) Schematic of model 1 of post-mitotic transcriptional bursting kinetics. (I,J) Fano factor for experimental and simulated data (using model 1). The shaded region represents the 95% confidence interval. (K) Schematic of model 2 of post-mitotic transcriptional bursting kinetics. (L,M) Fano factor for experimental and simulated data (using model 2). The shaded region represents the 95% confidence interval. *twi-DPE* N=3 embryos, *n*=357 nuclei; *twi-DPE+TATA* N=3 embryos, *n*=451 nuclei. See also Movies 7-9, Table S3 and the Materials and Methods.

bursting (see Materials and Methods, and supplementary Materials and Methods).

We considered two distinct models (Fig. 6H,K). Model 1 assumes that, after the first initiation event, transcription bursting parameters remain constant in time (Fig. 6H), while Model 2 introduces time inhomogeneity by allowing all the bursting parameters to evolve (Fig. 6K) (see Materials and Methods). We then ran numerical simulations for both models using parameters estimated from experimental data (*twi* enhancer transgenes). For Model 1, the steady state parameters are estimated in the interval 20-30 min after mitosis, a period identified as time homogeneous. For Model 2, the bursting parameters are considered to take different constant values over two distinct temporal windows: $t<20$ min and $20$ min$<t<30$ min. The parameters for $t<20$ min were estimated using the experimental data between the first postmitotic initiation and 20 min, whereas the parameters for $20$ min$<t<30$ min were the same as those used for Model 1. While the bursting parameters are not strictly constant within the window $t<20$ min, Model 2 provides a simple approximation of their time dependence.

We next applied these models to predict the time-dependent Fano factor observed in the *twi-DPE+TATA* and *twi-DPE* data, using the procedure explained above. As shown in Fig. 6I,J, Model 1 accurately reproduces the steady-state value of the Fano factor for both genotypes. However, despite incorporating a stochastic postmitotic lag, it does not produce the peak in transcriptional noise observed in the experimental data upon mitotic exit. This failure indicates that the postmitotic stochastic reactivation timings included in Model 1 are not sufficient to account for the presence of a peak in the transcriptional noise.

In contrast, Model 2 successfully reproduces the steady-state value and generates a peak of the Fano factor (Fig. 6L,M). This suggests that a postmitotic lag, followed by time-inhomogeneous bursting, may be required to observe a peak in the Fano factor.

The modeling approach considers only intrinsic, i.e. uncorrelated, sources of noise, including variability in the mitotic lag. Under this hypothesis, we correctly predicted the value of Fano at large times, suggesting that extrinsic (correlated) noise sources make only a limited contribution to the overall variability of the signal. We tested this hypothesis directly by decomposing the noise into intrinsic and extrinsic noise (Elowitz et al., 2002). Our method further partitions extrinsic noise into an intracellular component and an environment-based component, named extracellular extrinsic noise (Topno et al., 2025 preprint). The intracellular extrinsic component arises from fluctuations shared by two alleles within the same nucleus but vary independently between nuclei, such as transcription factor concentrations. In contrast, the extracellular extrinsic component reflects fluctuations common to all alleles, regardless of whether they reside in the same nucleus. We applied this noise decomposition to homozygous MS2 movies, where we simultaneously monitored nascent transcription from two identical reporters (*sna-MS2* or *sna+INR-MS2*) (Fig. S6A,B, Movie 12). We found that the intracellular extrinsic component is small compared to the intrinsic contribution (Fig. S6C,D). We therefore conclude that the transcriptional noise measured in this study primarily reflects intrinsic noise.

Collectively, our mathematical modeling distinguishes two separate sources of transcriptional noise: the timings to resume transcription upon mitotic exit and the kinetics of promoter switching. Importantly it reveals that not only do promoter switching kinetics drive noise, but that the perturbation of these bursting kinetics by mitosis and time-dependent regulators adds an additional layer of variability.

## DISCUSSION

Genetically identical cells often display an important cell-cell variability in gene expression, notably at the transcriptional level (transcriptional noise) (Pal and Dhar, 2024; Urban and Johnston, 2018). Understanding where this variability comes from is a fundamental issue. Here, we employed quantitative live imaging in the early *Drosophila* embryo to dissect how the cis-regulatory code contributes to inter-nuclear variability in transcription during patterning. By focusing on model developmental loci required for dorso-ventral patterning, we revealed that transcriptional noise is time-dependent and peaks after mitosis (Fig. 7A). Moreover, we demonstrate that the presence of a TATA box significantly enhances transcriptional noise through the mitotic lag time but also via specific bursting parameters (Fig. 7B).

### The mitotic lag time represents an important source of transcriptional noise

Since transcription generally ceases during mitosis, cell cycle heterogeneity has been considered as an important source of gene expression noise. During zygotic genome activation in zebrafish embryos, cell cycle heterogeneity has been shown to contribute to transcriptional noise (Stapel et al., 2017). Because non-synchronous cell cycling is often observed in embryos, it is considered an important extrinsic source of noise. We sought to exploit the naturally synchronized early *Drosophila* embryonic cell cycles with simple and short cycles comprising only S and M phases. Our data reveal the existence of transcriptional noise, even when mitoses are synchronized. Upon mitotic exit, transcription is reactivated at different times in each nucleus, a timing we refer to as mitotic lag time. We found that differences in mitotic lag time represent an important source of transcriptional noise. The mitotic lag time comprises a period due to DNA replication, believed to occur early in S phase (Shermoen et al., 2010), and a stochastic component prior to the first transcriptional activation. The time of first transcriptional activation represents a source of noise in all genotypes and spatial domains examined. However, the level of this noise is highly dependent on the cis-regulatory logic. Indeed, various molecular mechanisms can affect this mitotic lag time, as, for example, the priming of enhancers by pioneer factors or mitotic bookmarking (Dufourt et al., 2018; Fernandes et al., 2022; Lagha et al., 2013; Yamada et al., 2019; Yokoshi et al., 2022). The modeling of the mitotic lag time suggests that slow transitions are often associated with a higher level of transcriptional noise. In some contexts, mitotic lag time displays transcriptional memory (Bellec et al., 2022; Dufourt et al., 2018; Ferraro et al., 2016). Indeed, the population of nuclei entering nc14 may have experienced different transcriptional histories in previous cycles, leading to a marked heterogeneity of mitotic lag time. However, like most of the memory effects, timing differences between nuclei disappear with time. This phenomenon may explain the transient nature of the noise produced by the mitotic lag time, peaking after mitosis and progressively decreasing during the rest of the cell cycle until further regulatory inputs are engaged.

Previous work in murine cell lines established that about half of the transcriptionally active genome exhibits a transcriptional 'spike' at the mitosis-G1 transition (Hsiung et al., 2016). This spike corresponds to an increased probability for a cell to be in a transcriptionally active 'on' state, rather than an increase in the number of nascent mRNAs produced during an 'on' period. The parallel between *Drosophila* early embryos and mammalian cultured cells is hard to draw given the differences in cell cycles, yet we note that this spike in transcription is mirrored by the time-dependent

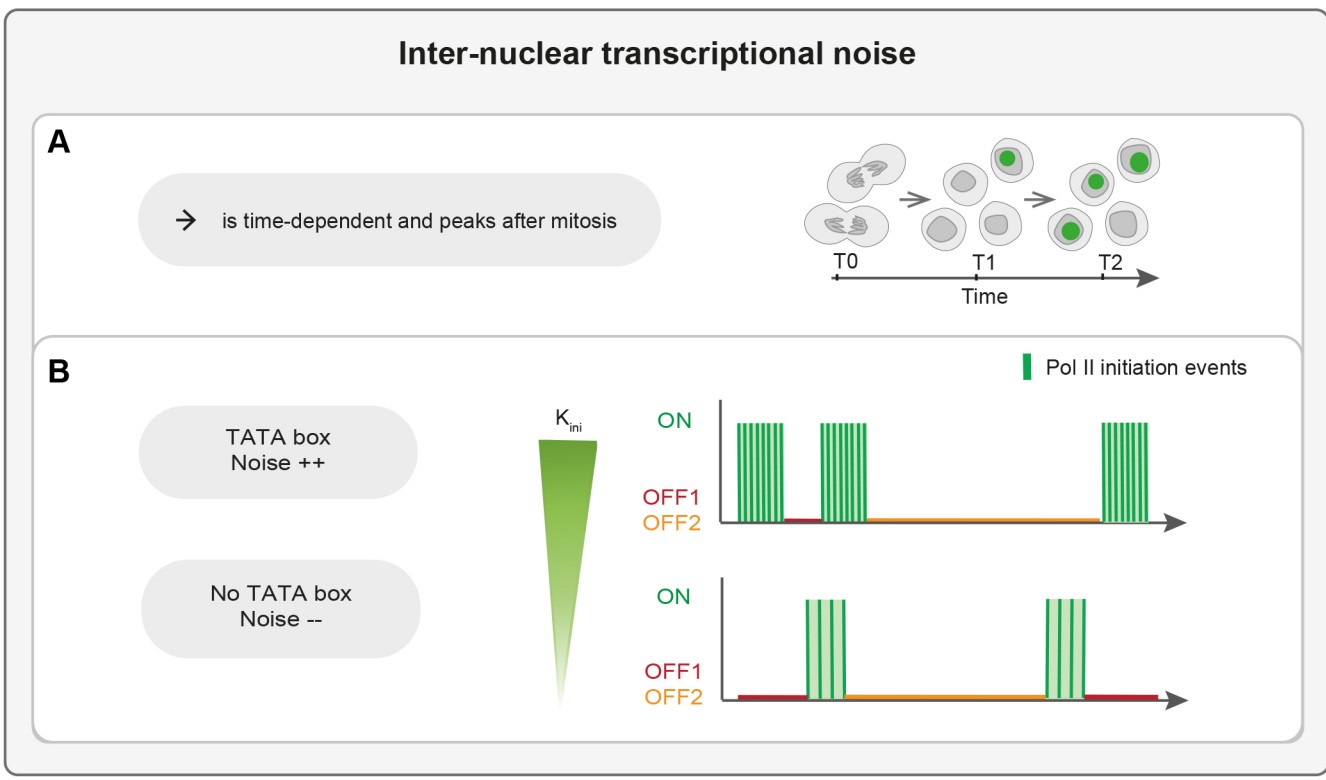

**Fig. 7. Schematic summarizing the main results of this study.** (A) Internuclear variability in Pol II initiation events is time dependent and peaks after mitosis. (B) The presence of a TATA box in a promoter is sufficient to elicit high transcriptional noise, in part due to its effect on the kinetics of transcriptional bursting.

bursting following mitotic lag time, which leads to the peak in transcription noise that we observe in *Drosophila* embryos. Given the prevalence of the mitosis-G1 transcriptional spike in vertebrates, we anticipate that our finding on the quantitative effect of mitotic lag time on noise in *Drosophila* embryos will inspire further investigation into the role of cell cycle progression on gene expression heterogeneity.

### The presence of multiple co-acting enhancers does not obviously contribute to transcription noise filtering

By quantifying noise from two model loci in various spatial domains, we show that two enhancers with overlapping activities do not buffer transcriptional noise more efficiently than when acting alone. This finding is counterintuitive, as redundant enhancers have been proposed to act as 'canalizers' of gene expression, especially under physiological or genetic stress (Kvon et al., 2021). It therefore seems plausible that transcriptional noise would be minimized in the context of multiple co-acting enhancers. Our results nuance this hypothesis and suggest that the capacity of shadow enhancers to suppress noise might be context dependent. To rigorously address this important question, a systematic approach is required, one that has been recently implemented in cultured mammalian cells (Tünnermann et al., 2025 preprint) but remains challenging to perform *in vivo*.

Two endogenous enhancers are never fully redundant and always differ in sequence, location and chromatin landscape. A recent study assessed how the duplication of a given enhancer behaves with respect to variability in comparison to a cognate pair of redundant/shadow enhancers (Waymack et al., 2020). Using a transgenic *Krüppel* (*Kr*) gene regulation paradigm *Drosophila* embryos, Waymack et al. showed that the *Kr* distal+proximal enhancer pair

produces lower expression noise than either of the duplicated enhancer pairs (proximal+proximal or distal+distal) (Waymack et al., 2020). They further conclude that shadow enhancer pairs achieve noise buffering via the separation of transcription factor input regulating each individual enhancer. We note that many distinct definitions of noise are used, as well as metrics to quantify them, which can be confusing. In the study of Waymack et al., gene expression noise was measured by calculating the coefficient of variation (CV, standard deviation/mean) of each transcription site with itself across nc14, then averaged between different embryos for each spatial position. Consequently, noise here refers to intra-nuclear fluctuations across time and space. In contrast to the noise captured in our measurements, which reflect inter-nuclear variability in transcriptional activity over time. Therefore, the aforementioned conclusions of Waymack et al. are not in contradiction with our findings but, instead, focus on another aspect of noise. Our ability to locate individual polymerase initiation events in living embryos allows us to grasp the very early steps of gene expression noise.

Finally, to be functionally relevant, transcriptional noise in nascent mRNA needs to 'flow' within the central dogma of molecular biology to lead to intercellular differences in protein products. Hence, beyond transcription, mRNA nuclear retention, diffusion and half-lives are known to strongly affect heterogeneity in cytoplasmic mRNA levels in mammalian cells (Battich et al., 2015; Baudrimont et al., 2019; Raj et al., 2006; Little et al., 2013). Future work considering these multiple regulatory layers operating at various stages of the mRNA lifecycle will provide insights into the propagation or filtering of nascent mRNA noise.

## Study limitations

Our study considers variability in the number of Pol II firing events between neighboring active nuclei, scored at various times and locations. We also note that our imaging-based study and conclusions are based on a small number of model loci.

Our study employs BurstDECONV to infer polymerase initiation events prior to calculating their inter-nuclear variability. This procedure assumes a processive Pol II, a constant Pol II speed and a rapid release of the nascent RNA. Pol II elongation speed is known to vary from gene to gene and was recently proposed to speed up within intronic sequences (Keller et al., 2023). Given the small number of enhancers examined in this study and their partially overlapping functions, we cannot rigorously investigate the effect of shadow enhancers on transcriptional noise.

## MATERIALS AND METHODS

### Fly husbandry

The *sna*, *sna$^{ΔDist}$* and *sna$^{ΔProx}$* BAC lines have been previously published (Bothma et al., 2015). The *sog*, *sog$^{ΔDist}$* and *sog$^{ΔProx}$* lines were a kind gift from Dr C. Rushlow (New York University, NY, USA) (Whitney et al., 2022). The *snaEnhancerCore* transgenes (Fig. 5A), composed of a fragment of the *sna$^{Dist}$* enhancer (*sna$^{DistCore}$*; Ferraro et al., 2016) followed by the indicated promoter, 24xMS2 repeats and the *yellow* sequence as a reporter, were previously published by Pimmett et al. (2021).

The *twi* transgene series was generated for this study (BestGene; described below). All crosses were maintained at 25°C. For live imaging, homozygous males carrying the MS2 reporter lines were crossed with homozygous virgin females bearing the *MCP-eGFP-His2Av-mRFP(III)* construct, except in the *dl$^1$/+* context where reporter males were crossed with *dl$^1$/+;;MCP-eGFP-His2Av-mRFP/+* virgin females.

### Transgenic line generation

pbPhi was digested with NotI and BamHI followed by gel purification. The *twist* upstream *cis*-regulatory region comprising the distal and proximal enhancers to 50 bp downstream of the transcriptional start site was amplified from sequences generously provided by Dr T. Juven-Gershon (Zehavi et al., 2014) and inserted into pbPhi using HiFi DNA Assembly master mix (New England Biolabs). The same region was amplified from plasmids containing promoter mutations (*twi-DPE* and *twi-DPE+TATA*) and inserted into pBPhi using the same strategy. Plasmids were sequenced to ensure accuracy prior to PhiC31-mediated integration into VK33 (BL 9750) by BestGene.

Primers used for amplification were: forward, ttacggaacgcctgtgacgcaggatccttccggcc; and reverse, ggccggaaggatcctgcgtcacaggcgttccgtaa

### Live imaging

Embryos were collected after 3 h of egg laying at 25°C. Embryos were manually dechorionated using double-sided tape and oriented with the dorsal side down on a hydrophobic membrane supplemented with heptane glue and covered with immersion oil 10S (VWR Chemicals) and a coverslip. Embryos were imaged using an LSM 880 with a Fast Airyscan module and the Zen Black software (Zeiss). MCP-GFP and His2av-mRFP excitation by a 488 nm and 561 nm laser, respectively, were captured on a GaAsP-PMT array with an Airyscan detector using a 40× Plan Apo oil lens (NA=1.3). Movies were acquired over a *z* volume of 14.5 μm using 30 *z*-slices spaced at 0.5 μm, ensuring all MCP-GFP signal was captured within the volume. Laser power for each movie was verified using a ThorLabs PM100 optical power meter (ThorLabs) with the 10× objective at load position. Laser power for the 488 nm line was determined empirically based on genotype to avoid saturation and kept constant within an experimental series. The 561 nm laser was maintained at 17.5 μW across all samples. Experimental series-specific acquisition parameters were as follows. For *snail* BAC lines: zoom 2.5, resolution 512×512 pixels with bidirectional scanning, 488 nm laser intensity is 8 μW and time resolution is 4.64 s/*z*-stack. For *sog* CRISPR lines: zoom 2, resolution 800×800 pixels with bidirectional scanning, 488 nm laser intensity is 4.9 μW and time resolution is

6.35 s/*z*-stack. For *snail* enhancer and promoter (with *dl$^1$/+* background) transgenic lines: zoom 3, resolution 512×512 pixels with bidirectional scanning, 488 nm laser intensity is 10.5 μW and time resolution is 3.86 s/*z*-stack. For *twist* transgenic lines: zoom 2, resolution 800×800 pixels with bidirectional scanning, 488 nm laser intensity is 8.5 μW and time resolution is 6.35 s/*z*-stack. Data acquired from transgenes except the *twi* series and the *dl$^1$/+* background series are derived from Pimmett et al. (2021).

### Image analysis for live imaging

The transcription site intensity was tracked in 3D across time using SegmentTrack_v4.0 (Pimmett et al., 2021) available at https://github.com/ant-trullo/SegmentTrack_v4.0. Nuclei were spatially selected to maintain a common underlying gene regulatory network within an experimental series. For the *sna* BAC lines and *snaEnhancerCore* transgenes in wild type and *dl$^1$/+* contexts, nuclei within ±25 μm of the ventral furrow were selected. For *twi* transgenes, nuclei within ±35 μm of the ventral furrow were retained. For *sog* lines, selection was based on expression analysis. The mesoderm-neuroectoderm boundary is identified by the repression of *sog* in the mesoderm in mid-nc14. From this boundary, 30 μm zones are defined moving dorsally to delineate the ventro-lateral, dorsal-lateral and dorsal areas, and the ventral region in the presumptive mesoderm. If *sog* repression in the mesoderm is not observable, the marker is placed at the dorsal repression boundary, and 30 μm zones are defined moving ventrally from this point. Only the transcriptional activity within these regions was retained for all subsequent analysis steps.

The spot detection algorithm on rare occasions identified sister chromatids as separate transcriptional sites within a nucleus, confounding detection. To correct for this, an additional post-processing tool was developed (https://github.com/ant-trullo/SpotsFiltersTool; Pimmett et al., 2025). This tool introduced a specific parameter, defined as the ratio between the convex hull surface formed by the two spots and their actual size. Sister chromatids were identified and regrouped based on an empirically determined threshold for this ratio, as the chromatids were expected to remain relatively close in space, while objects appearing as false detections are not spatially restricted. To prevent some blinking activations from being discarded, spots that disappeared for a certain number of frames, which were defined based on the length from the beginning of the MS2 cassette to the end of the transcriptional unit, were retained based on a user-defined threshold process. Nuclei not exhibiting any transcriptional activation during the temporal analysis window were discarded. In all cases, the first detection for calculation of the activation percentage was marked at the first time point where a transcription site was detected and retained based on the filtering described above. After sister chromatid correction, false detection filtering and spatial tracking, intensity traces were calibrated as described by Pimmett et al. (2021).

To facilitate the analysis of data with two labelled alleles, we developed a post-processing tool (https://github.com/ant-trullo/SistersSplitTool) as previously described (Pimmett et al., 2025). Briefly, as a given transcription site is (1) largely spatially restricted to a small local area and because (2) labelled alleles overlap very rarely, an algorithm was developed to independently track each transcription site in a given nucleus. At each time point, the position of each allele is recorded with respect to the center of mass of the nucleus to account for nuclear drift. Plotting these coordinates yields two-point clouds with limited overlap. We resolve them by fitting the distribution with a two-component Gaussian mixture model. The fit is then used to assign an identity to each transcription site within the nucleus, with manual correction performed when necessary.

### Live experiment noise calculation

For all data, the period of analysis was limited from the end of mitosis/beginning of telophase until 30 min after mitosis. Following calibration, the signal was deconvolved into Pol II initiation events using BurstDECONV (Tantale et al., 2021; Douaihy et al., 2023) using the parameters in Table 1, with a Pol II speed of 25 bp/s, a mRNA retention time of 0 s and a minimum Pol II distance of 30 bp. BurstDECONV is publicly available at https://github.com/oradules/BurstDECONV.

**Table 1. BurstDECONV hyperparameters**

| Genotype | sog CRISPR lines | sna BAC transgenic lines | sna transgenic lines | twi transgenic lines |
|---|---|---|---|---|
| mRNA length before MS2 sequence | 2553 bp | 146 bp | 41 bp | 54 bp |
| MS2 length | 1267 bp | 1292 bp | 1292 bp | 1292 bp |
| mRNA length after MS2 sequence | 9081 bp | 5467 bp | 4526 bp | 4526 bp |

### Fano factor calculation of Pol II initiation events from mitosis

Following deconvolution, the number of Pol II initiation events was calculated for each individual nucleus within the full length of the movie. We then extract the number of polymerases contributing to the signal in a time window equal to the dwell time (i.e. the time it takes for a polymerase to transcribe a single mRNA). This window provides a 'snapshot' perspective of all polymerases contributing to the signal intensity at time $t$.

The Fano factor at time $t$ is calculated by determining the variance and the mean of the number of Pol II initiation events across the pooled nuclei within that time window. This process is then repeated every 1 min along the length of the movie to extract the Fano factor across time. Thus, we can plot the Fano factor of Pol II initiation events for a population of nuclei over time, using a sliding window of dwell time length specific to the mRNA transcript type, updated every minute. We used the following dwell time values: 413.92 s for *sog* CRISPR lines, 270.36 s for *sna* BAC transgenic lines, and 232.72 s for *sna* and *twi* transgenic lines. The Fano factor calculation code is available at https://github.com/mariadouaihy/transcriptional_bursting_noise.

### Fano factor calculation of Pol II initiation events after mitotic lag times elimination

To eliminate the component of noise caused by the time each nucleus takes to activate or reactivate after mitosis (mitotic lag time), the time origin was individually shifted to the first point at which transcriptional activity reached one-fifth of the maximum intensity for each nucleus. The one-fifth threshold was defined to exclude early false detection events (Pimmett et al., 2021). The subsequent calculation of Fano factor across time was then performed as described in the previous section.

### Modeling the relationship between steady state Fano factor and bursting parameters

The times of successive initiation events generated by a bursting promoter can be modeled as a renewal process (Radulescu et al., 2025, see supplementary Materials and Methods). Renewal processes generalize the Poisson process by considering that the waiting time between successive events is not necessarily exponentially distributed. However, as for Poisson processes, successive waiting times are considered independent. Furthermore, it can be shown (Feller, 1968) that both variance and the mean of the number of events $N$ are proportional to the window length $T$, if $T$ is large enough:

$$Var(N) = \frac{T\,\sigma^2}{\mu^3}, \quad \langle N \rangle = \frac{T}{\mu}, \tag{1}$$

where $\sigma$, $\mu$ are the standard deviation and the mean of the waiting time between successive events (for a Poisson process, the waiting time is exponentially distributed, so $\sigma = \mu$, but in general $\sigma$, $\mu$ are independent).

From Eqn 1 we get

$$Fano(N) = \frac{Var(N)}{\langle N \rangle} = \left(\frac{\sigma}{\mu}\right)^2, \tag{2}$$

which shows that the Fano factor does not depend on the window length if this is large enough.

We have tested this property empirically and, indeed, for windows lengths $T > 1$ min, Fano does not depend on $T$.

The mean waiting time is also related to the product $P_{ON} k_{ini}$, where $P_{ON}$ is the probability that the promoter is ON and transcribing, and $k_{ini}$ is the transcription initiation rate in this state (we consider that there is no

transcription in the OFF state). At stationarity, the number of initiation events $N$ on a very large time interval $T$ can be computed in two ways

$$N = \frac{T}{\mu} = P_{ON}\, k_{ini}\, T. \tag{3}$$

It follows from Eqn 3 that:

$$P_{ON}\, k_{ini} = \frac{1}{\mu}. \tag{4}$$

According to Eqn 4, we can estimate $P_{ON}\, k_{ini}$ by estimating $\mu$. The estimation of $\mu$ is performed by counting all the transcription initiation events, for all the nuclei that are contained in a time window of length $T$. Then, we compute the mean of the waiting times separating successive events as $\mu = \langle \frac{T}{N_w} \rangle$, where $N_w$ is the number of events in the window for a nucleus, and the mean is over the nuclei.

Finally, we were able to relate the transcription noise parameters $\sigma$, $\mu$ and *Fano* to the bursting transition rates between the ON and OFF states. This is important because it allows us to understand the origin of quantitative differences in noise.

In order to do so, we proceeded in two steps. For a bursting model with $N$ states, the distribution of the waiting time between successive initiation events is multiexponential and characterized by the survival function

$$S(t) = P(\tau > t) = A_1\, exp^{\lambda_1\, t} + \cdots + A_N\, exp^{\lambda_N\, t}, \tag{5}$$

where $A_1 + \cdots + A_N = 1$ and $\lambda_i < 0$, $i = 1, \ldots, N$ (Tantale et al., 2021; Radulescu et al., 2025).

One has the following relations (see Tantale et al., 2021):

$$\mu = -\left(\frac{A_1}{\lambda_1} + \cdots + \frac{A_N}{\lambda_N}\right) \tag{6}$$

$$\sigma^2 = 2\left(\frac{A_1}{(\lambda_1)^2} + \cdots + \frac{A_N}{(\lambda_N)^2}\right) - \mu^2. \tag{7}$$

The multiexponential parameters $A_i$ and $\lambda_i$, for $i = 1, \ldots, N$, result from the BurstDECONV fit with $N$ states ($N$ exponentials) and allow the computation of the Fano factor using Eqns 6, 7 and 2. However, we would like to know how the Fano factor depends on the bursting rates.

The multiexponential parameters $A_i$ and $\lambda_i$, for $i = 1, \ldots, N$, can be related to the bursting parameters. The mappings from multi exponential parameters to bursting parameters and back can be computed automatically, and some solutions for promoter models with two, three and four states are listed in Radulescu et al. (2025) and Tantale et al. (2021).

For the sake of completeness, we provide analytical formulas relating the bursting parameters to the steady-state Fano for two models: the telegraph, the two state model and the three-state model illustrated in Fig. 6D.

Using the formalism of Radulescu et al. (2025) and Eqns 2, 6 and 7, we derive, for the three states model

$$FANO = 1 + \frac{2k_{ini}(k_2^{off}\,(k_1^{on})^2 + k_1^{off}\,(k_2^{on})^2)}{(k_1^{on} k_2^{on} + k_1^{on} k_2^{off} + k_2^{on} k_1^{off})^2}. \tag{8}$$

It is useful to also have the mean transcription intensity, which is the inverse of $\mu$ (see Eqn 4):

$$P_{ON} k_{ini} = \frac{k_1^{on} k_2^{on} k_{ini}}{k_1^{on} k_2^{on} + k_1^{on} k_2^{off} + k_2^{on} k_1^{off}}. \tag{9}$$

For the two state model, we have

$$FANO = 1 + \frac{2k_{ini} k_{off}}{(k_{off} + k_{on})^2} \tag{10}$$

and

$$P_{ON}\, k_{ini} = \frac{k_{on} k_{ini}}{k_{on} + k_{off}}. \tag{11}$$

Eqns 10 and 11 show that Fano and the mean transcription intensity are generally independent. Indeed one can keep the ratio $k_{on}/k_{off}$ and $k_{ini}$ fixed

but slow down the state transition rates (reduce $k_{on}$ and $k_{off}$) to increase Fano. Similarly, one can multiply $k_{ini}$, $k_{on}$ and $k_{off}$ by the same factor $r$ (which means keep the ratios $k_{ini}/k_{on}$, $k_{on}/k_{off}$ fixed). This leaves Fano invariant but multiplies $P_{ON} k_{ini}$ by $r$.

Eqns 8 and 9 lead to the same conclusion. The general justification for the independence is that the Fano factor minus one and the mean intensity are homogeneous rational functions (i.e. ratios of homogeneous polynomials, where each polynomial consists of monomials of the same degree in the parameters), but the degrees of homogeneity – the difference between the degrees of the numerator and denominator – are not the same for the two functions.

### Modeling the relation between the Fano factor and the postmitotic lag

The statistics of the postmitotic lag $T_0$ has been modeled as $T_0=T_d+T_s$, where $T_d$ is an incompressible deterministic delay and $T_s$ follows a mixed Erlang distribution, as in Dufourt et al. (2018) and Bellec et al. (2022). This distribution corresponds to a 'staircase' model of reactivation that considers that nuclei may need *1, 2, …, M* successive exponentially distributed steps, each of a mean duration $\tau$ to activate. Such a distribution is characterized by the parameters $p_1, p_2, …, p_M$ and $\tau$, representing the probabilities that a nucleus needs *1, 2, …, M* steps to activate and the mean duration of each step, respectively.

More precisely, the stochastic component $T_s$ is given by a mixture of Erlang distributions with shape parameters *1, 2, 3,…* and scale parameter $\tau$, the cumulative distribution function (cdf) of which reads:

$$F(t) = P\ [T_s \leq t] = p_1 \left( 1 - exp^{\frac{-t}{\tau}} \right) + p_2 \frac{1}{\Gamma(2)} \gamma\left(2, \frac{t}{\tau}\right)$$
$$+ p_3 \frac{1}{\Gamma(2)} \gamma\left(3, \frac{t}{\tau}\right) + \cdots, \qquad (12)$$

where $\Gamma$ and $\gamma$, are the complete and incomplete gamma functions, respectively.

The parameters of the mixed Erlang distribution are estimated using the method introduced by Dufourt et al. (2018). The maximal number of steps $M$ is determined using a parsimony principle: it is the smallest $M$ for which the predicted survival function of the activation time is within the confidence interval of the empirical survival function (computed using Kaplan–Meier's method and Greenwood's formula for variance). The remaining parameters are obtained by minimizing an objective function based on the least squares differences between empirical and predicted survival functions. Together with the objective function value, the confidence interval test is a way to test the quality of the fit. The results showing the selection of $M$ and the quality of the fit are illustrated in Fig. S7.

### Simulated sequence of initiation events (models 1 and 2)

We used simulations to generate artificial data consisting of sequences of transcription initiation events. The data are represented as an increasing sequence of times:

$$0 < T_0 < T_1 < \cdots < T_N < t_{\max},$$

where $T_0$ is the first initiation event and $T_N$ is the last initiation event before the movie length $t_{\max}$.

Generation of $T_0$ and subsequent events was as follows. $T_0$ was generated using a mixed Erlang distribution, with the best-fit parameters corresponding to the postmitotic reaction gap observed in experimental data, and a deterministic delay given by the minimum reactivation time across nuclei. The times $T_i$ with $i>0$ were generated such that $T_{i+1}-T_i$ are distributed according to a three-exponential distribution (corresponding to a three states bursting model). The parameters of the three exponential distributions are $A_1, A_2, A_3, \lambda_1, \lambda_2$ and $\lambda_3$, and were chosen differently for the two models:

### Model definitions

Model 1
The parameters remain constant and are set to the values extracted from experimental data for times between 20 and 30 min.

Model 2
The parameters are piecewise constant, as follows. If $T_i<20$ min, we use the best fit parameters from experimental data for times smaller than 20 min, excluding the postmitotic gap. If 20 min$<T_i<30$ min, we use the best-fit parameters from experimental data for times between 20 and 30 min. The code for modeling the Fano factor is available at https://github.com/oradules/figure6_maillard_et_al.

### Total variance decomposition to obtain intracellular and extracellular extrinsic noise components

Here, we follow the methodology introduced by Topno et al. (2025 preprint). We employ a two-reporter system with two equivalent alleles per nucleus and record the MS2 signal from both alleles simultaneously (see Fig. S6). For the analysis, we perform pairwise comparisons of the signals. Sites within the same nucleus are referred to as paired, whereas sites selected randomly from distinct nuclei are referred to as unpaired.

The variance of the MS2 signal can be decomposed into three components using the law of total variance:

$$Var_{total} = Var_{intrinsic} + Var_{extrinsic-intra} + Var_{extrinsic-extra}. \qquad (13)$$

Intrinsic noise corresponds to the uncorrelated component of variance, arising from stochastic events that independently affect each allele. Extrinsic noise corresponds to the correlated component of variance, which can be further divided into two contributions. Intracellular extrinsic noise arises from fluctuations that are shared by two alleles within the same nucleus but vary independently between nuclei. Extracellular extrinsic noise arises from fluctuations that are common to all alleles, regardless of whether they are paired or unpaired. The different components are estimated as follows:

$$Var_{intrinsic} = \frac{1}{npaired + nunpaired} \left( \sum_{i=1}^{npaired+nunpaired} \frac{1}{2}(x_i^t - y_i^t)^2 \right) \quad (14)$$

$$Var_{extrinsic-intra} + Var_{extrinsic-extra} = \frac{1}{npaired} \sum_{i=1}^{npaired} x_i^t \cdot y_i^t - \overline{x}^{\,t} \cdot \overline{y}^{\,t} \quad (15)$$

$$Var_{extrinsic-extra} = \frac{1}{nunpaired} \sum_{i=1}^{nunpaired} x_i^t \cdot y_i^t - \overline{x}^{\,t} \cdot \overline{y}^{\,t}, \qquad (16)$$

where $x_i^t$, $y_i^t$ are values of mRNA numbers for pairs of transcription sites (TSs) in frames corresponding to a small window centered at time $t$, and $\overline{x}^{\,t}, \overline{y}^{\,t}$ are the respective means. In Eqn 14 the sum is over both true pairs (same nuclei TS, *npaired*=number of nuclei $N$) and random pairs (TS from different nuclei, $nunpaired \leq N(N-1)/2$). In Eqn 15, the sum is over true pairs, whereas in Eqn 16, the sum is over random pairs. The Matlab implementation of the variance decomposition code is available at https://github.com/oradules/extrinsic_intrinsic_decomposition_snail.

### Acknowledgements

We are grateful to Tamar Juven-Gershon for sharing the *twist* promoter plasmids. We thank Christine Rushlow for generating and sharing the *sogMS2* stocks. We thank Edouard Bertrand and the LabMuse network for insightful discussions. We are grateful to Amandine Palandri for help with fly handling. We thank all members of the Lagha lab for critical reading of the manuscript. We acknowledge the Montpellier Resources Imagerie facility (France-BioImaging) for imaging and the Biocampus *Drosophila* facility of Montpellier for fly food.

### Competing interests

The authors declare no competing or financial interests.

### Author contributions

Conceptualization: M.L., O.R.; Data curation: L.M., V.L.P., M.D., A.T.; Formal analysis: V.L.P., M.D., R.T., O.R.; Funding acquisition: M.L.; Investigation: M.L., L.M., V.L.P., M.D., P.G.-I., A.B., A.T., O.R.; Methodology: L.M., M.D., O.R.; Project administration: M.L.; Software: M.D., A.T., O.R.; Supervision: M.L., O.R.; Validation: M.L., L.M., V.L.P., O.R.; Visualization: L.M., V.L.P., O.R.; Writing – original draft: M.L., L.M., O.R.; Writing – review & editing: M.L., L.M., V.L.P., M.D., P.G.-I., A.B., A.T.

## Funding

Research in the laboratory of O.R. is supported by the Université de Montpellier, the Centre National de la Recherche Scientifique and the Institut National de la Santé et de la Recherche Médicale. Research in the laboratory of M.L. is supported by the Centre National de la Recherche Scientifique and the Université de Montpellier, and is funded by grants from the Fondation Bettencourt Schueller, the European Molecular Biology Organization Young Investigator program (EMBO YIP), the European Research Council (SyncDev, LightRNA2Prot) and Agence Nationale de la Recherche HubDyn. M.D. was supported by the Center National de la Recherche Scientifique and a University of Chicago Joint PhD program. L.M. and R.T. were supported by a fellowship from the LabMuse/Université de Montpellier PhD program, and by PhD fellowships from La Ligue Contre Le Cancer (to L.M.) and Sidaction (to R.T.). M.L., O.R. and A.T. are sponsored by the Center National de la Recherche Scientifique. Open Access funding provided by M.L. Deposited in PMC for immediate release.

## Data and resource availability

All relevant data and details of resources can be found within the article and its supplementary information.

## Peer review history

The peer review history is available online at https://journals.biologists.com/dev/lookup/doi/10.1242/dev.204953.reviewer-comments.pdf

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
