## [Peer Review File · Development (Cambridge, England)]

Mitotic reactivation and transcriptional bursting govern transcriptional noise in the early *Drosophila* embryo

Louise Maillard, Virginia L. Pimmett, Maria Douaihy, Pablo Garcia-Idieder, Rachel Topno, Amelie Brun, Antonio Trullo, Ovidiu Radulescu and Mounia Lagha
DOI: 10.1242/dev.204953

Editor: James Briscoe

Review timeline

Original submission:	20 May 2025
Editorial decision:	29 June 2025
First revision received:	4 February 2026
Accepted:	26 February 2026

Original submission

First decision letter

MS ID#: dev.204953

MS TITLE: Mitotic reactivation and transcriptional bursting govern transcriptional noise in the early *Drosophila* embryo

AUTHORS: Mounia Lagha, Louise Maillard, Virginia L. Pimmett, Maria Douaihy, Pablo Garcia Idieder, Amelie Brun, Antonio Trullo and Ovidiu Radulescu

Dear Dr Lagha,

I have now received all the referees' reports on the above manuscript, and have reached a decision. The referees' comments are appended below, or you can access them online: please go to:

As you will see, the referees express considerable interest in your work, but have some significant criticisms and recommend a substantial revision of your manuscript before we can consider publication. Specifically, all three reviewers question whether the main finding, that redundant enhancers modulate transcriptional noise, is adequately supported by the data. The reviewers point out that the enhancers you studied exhibit distinct and sometimes opposing behaviours, with the clearest example of truly redundant enhancers (the sna series) showing minimal differences in noise levels. Furthermore, the reviewers raise the possibility that the observed changes in Fano Factor may reflect differences in burst strength that correlate with expression levels, rather than representing modulation of transcriptional noise. To address these concerns, the reviewers recommend further analysis of the data, including inferring the underlying bursting parameters across all experimental conditions.

If you are able to revise the manuscript along the lines suggested, which may involve further experiments, I will be happy receive a revised version of the manuscript. Your revised paper will be re-reviewed by one or more of the original referees, and acceptance of your manuscript will depend on your addressing satisfactorily the reviewers' major concerns. Please also note that Development will normally permit only one round of major revision. If it would be helpful, you are welcome to contact us to discuss your revision in greater detail. Please send us a point-by-point response indicating your plans for addressing the referees' comments, and we will look over this and provide further guidance.

Please attend to all of the reviewers' comments and ensure that you clearly highlight all changes made in the revised manuscript. Please avoid using 'Tracked changes' in Word files as these are lost in PDF conversion. I should be grateful if you would also provide a point-by-point response detailing how you have dealt with the points raised by the reviewers in the 'Response to Reviewers' box. If you do not agree with any of their criticisms or suggestions please explain clearly why this is so.

Reviewer 1

Advance summary and potential significance to field

Maillard et al provide detailed quantifications of transcriptional dynamics (MS2-MCP system) for two developmentally important genes (*sna* and *sog*) across a series of CRE deletions. The paper's strengths are the careful live imaging, and detailed temporal analysis. In principle, comparisons of the contributions of "redundant"/"shadow" enhancers to transcription and associated noise is important to understand how multiple regulatory inputs are integrated to achieve cell fate decisions during development, and to do so robustly. Efforts to combine detailed measurement with quantitative modelling is key to solve this problem, and is something that the authors are helping contribute towards.

Comments for the author

Overall, the manuscript suggests that redundant enhancers modulate transcriptional noise, but this conclusion is not fully supported by the data, as the enhancers studied show differing behaviours and the clearest redundant pair shows minimal noise differences. Additionally, observed changes in Fano Factor may largely reflect differences in burst strength linked to expression level, rather than direct modulation of transcriptional noise. Before publication, the following major points should be addressed:

1. Misinterpretation of Fano Factor (FF) and Noise

Fano Factor appears to scale with mean expression, which is likely due to increased burst strength. This is distinct from a true modulation of transcriptional noise.

The authors claim opposing behaviour in *sog* vs *sna*, but these differences if any are not clear at all, given the above: in both cases, higher-expressing constructs show FF peaks, as expected if burst strength increases.

2. Misuse of the Term "Redundant Enhancers"

The enhancers examined show distinct and sometimes opposing regulatory behaviour, with one (*sog* proximal) potentially acting as a repressor, undermining the claim of redundancy.

The *sna* enhancer series—the only true redundant pair—shows no difference in noise, which contradicts the main claim that redundant enhancers modulate transcriptional noise.

3. Lack of Inferred Bursting Parameters

The authors should systematically infer burst frequency, strength, initiation intensity, and duration from their data.

They only attempt this for one promoter (TATA box example), even though this analysis is critical to properly interpret all experiments.

Without these parameters, it's unclear whether the Fano Factor differences actually reflect modulation in transcriptional noise, rather than some burst correlate of mean expression.

4. Missing or Incomplete Data Presentation

Mean expression traces are missing or relegated to the supplement, making it challenging/impossible to interpret the FF plots.

All aligned analyses need to show the corresponding mean expression and aligned time traces. Likewise for the TATA box analysis

5. Overstated and Unsupported Conclusions

- The main conclusion—that noise differs between redundant enhancers—is not supported: those that differ appear not to be redundant; those that are redundant appear show no difference.

Additionally, some further points should be responded to:

The paper presents two possible interpretations of transcriptional noise: (a) intrinsic, internal fluctuations at a single locus over time, and (b) inter-nuclear or population-level variability measured across nuclei within time windows. The apparent reliance on inter-nuclear variability could be inconsistent with models based on intrinsic noise. The authors should clearly define which type of noise they are measuring, what assumptions they're making therein, justify its use in the context of their modelling, and, if necessary, re-analyse their data to distinguish between intrinsic and population-level noise sources.

The manuscript could benefit from clearer presentation, both in the text and figures. Axis scales are missing in places, and formatting at times is inconsistent, and there are plenty of typos.

Line 159-160: Figure 2D plots the absolute intensity of transcripts. It is therefore immediately obvious to distinguish which allele has "more variable transcription". Reference to the "y-axis" here doesn't help much.

In Figure 2C-D, it appears that Δ Prox shows a higher peak of expression than the WT. Can the authors comment somewhere whether this proximal element is thus functioning as a (partial) silencer? Whether these sequences have positive or negative influences on transcription alters the interpretation of noise regulation.

Line 185: "the control" should be described as sna^{WT}

Lines 195-201: This paragraph lacks any references to data. It is unclear whether this is new observations, re-interpretations of old ones (which would be quite a stretch), or speculation.

A more detailed description of how the Erlang model is fit is warranted. For presenting results, consider presenting something more than a schematic (Fig. 3J), for example providing evidence for model fit, and some sense of inter-embryo variability in measurements. It appears that using Erlang model makes sense only if considering temporal noise of a single cell, or a reasonable proxy thereof (see above).

The authors should discuss their work in relation to "Alamos, S., Reimer, A., Westrum, C., Turner, M. A., Talledo, P., Zhao, J., Luu, E., & Garcia, H. G. (2023). Minimal synthetic enhancers reveal control of the probability of transcriptional engagement and its timing by a morphogen gradient. *Cell Systems*, 14(3), 220-236.e3. <https://doi.org/10.1016/j.cels.2022.12.008>", which also considers Dorsal response, transcriptional induction, multi-step transcriptional initiation and mitotic entry contributions.

Methods appear to be missing for how "Cumulative activation percentage (mean \pm s.e.m. over embryos) by genotype" is calculated. Does "activation" imply intensity above some low fixed threshold, or is the threshold variable, or is it something else. Is this the proposal of "the one-fifth threshold" (Line 888)? If so, can the authors appreciate how altered time-scale of activation in various mutants could be undermined/mis-interpreted due to different maximum values?

"This suggests that the necessary and sufficient condition for observing a peak in the Fano factor is to have a postmitotic lag, followed by time-inhomogeneous bursting." This extremely strong claim is not fully warranted. The modelling suggests its sufficient, but there are likely many possible mechanisms (Lines 405-406)

In Table S3, the confidence intervals are remarkably tight (identical to 4-5 significant figures). Are these those that come out of the BurstDECONV software. It would be good to have a brief description of how to interpret these. Did the authors try fitting these separately by embryo, or via bootstrapping, to get a sense of the real variance.

Minor comments:

stray comma in Line 76

Lines 91-92: Appreciate the more general description of "redundant enhancers", but readers may benefit from their historical and perhaps better known descriptor "shadow enhancers".

Line 96: "fluctuations on..." should probably be "fluctuations in..."

Line 133: stray comma in "Transcriptional bursting, results"

Line 142: "Dorsal and are essential ..." typo.

Figure 1C bottom. The "n" in "tn" should be subscripted for consistency.

Line 178: Is there need to spell out "sna delta distal/proximal"? Especially given this is not provided for the sog alleles.

Line 210: sub-clause "particularly the enhancer" needs re-writing for improved clarity.

Line 213: Missing space "bookmarking(Bellec et al., 2022 ...".

X-ticks are mixing in Fig 2D and 2I.

Figure 3: it would help to have the *sog* on the left and *sna* on the right for consistency with Figure 2. It should be cited somewhere that Fig3B and G are repeated from Figure 2.

Given repeated data, is there a reason to show different y-axis extents for Fig 3B and Fig 2L.

Raw intensities in the various reporters analysed in Figure 5 should be reported somewhere, so readers can distinguish intensity variations from noise.

Heat map in Figure 6 should use stronger colours (e.g. Greyscale) so that the "white intervals" referred to in the text are visible.

Models 1 and 2 in Figure 6 should have corresponding schematics so readers can understand their differences clearly.

Methods: missing space in "The mappings from multiexponential" (line 935)

Missing close bracket in Line 830.

Reviewer 2

Advance summary and potential significance to field

The paper sets out to investigate sources of transcriptional noise and how they might contribute to developmental fate decisions. They utilize the elegant live-imaging approaches for which the Lagha lab is well-known and combine that with sophisticated analysis and modeling to tease out the underlying principles.

Strengths include a thoughtful description of transcription noise, decoupling changes in intensity from noise and inter-nuclear variability. They also apply a consistent parameter of noise (F-Factor) throughout, using it to measure noise across scales of temporal, cell-cycle and spatial based variabilities. The results highlight that the heterogeneity in the reinitiation of transcription following exit from mitosis is one major source of transcriptional noise. Notably, the coordinated cell divisions in the *Drosophila* embryo provided a good vehicle to detect this feature, which would be hard to distinguish in heterogeneous populations. They discuss the factors that might influence this aspect of noise ("mitotic lag") but they don't consider the potential impact for cell fate decisions- might this contribute to a period when cells are more plastic after mitosis for example?

The authors also investigate contributions from promoter elements, turning to defined synthetic reporters in place of the engineered gene loci. Their results that promoter elements impact differently on the levels of transcriptional noise and the final models suggest that the promoter kinetics are one factor influencing noise.

The results are interesting and the analysis informative, especially with respect to the mitotic lag. There are some concerns about the mixture of genetic tools that have been used and whether these might confound the interpretations. Nevertheless, the results bring valuable insights. For those to be accessible to a wide readership, the authors are recommended to clarify the text and the arguments as in the suggestions below.

Comments for the author

The results are quite confusing for a reader who is not deeply familiar with these types of data and analysis. In some places very little explanations are given, and the reader has to refer to previous papers to understand the methods. They are also quite selective in the results that they emphasize, and it is not always clear why.

The *sog* Δ prox has the most distinctive behaviour in all of the assays. This variant behaves very differently from all the others tested (but more similar to the synthetic reporters?) Is there a concern that this is behaving anomalously - Could the hyperactivity of the *sog* Δ prox be a consequence of the deletion being within the intron, impacting for e.g. on RNA stability?

In the initial experiments they state that *sog* Δ prox "shows more variability" than *sog*WT or *sog* Δ dist referring to Figure 2D. The graphs there have very different scales, but as presented. the *sog* Δ prox seems to have the least variability (highest intensity but fairly smooth traces). Better presentation

of the data is needed to clarify this point. (these graphs are very pixelated and vary in scales, presentation could be improved). The changes (decrease in intensity and F-factor) in both $\text{sog}\Delta\text{Dist}$ and $\text{sna}\Delta\text{Dist}$ (Fig 2C,J,F,L) are not discussed or elaborated on.

One of their initial conclusions is that the intrinsic noise is not related to enhancer strength. But the effects from deleting sna enhancers on levels of expression (the measure of strength) are relatively minor unlike those with the sog where there is a large-scale change. Don't the results suggest that normally the sog -prox enhancer helps to suppress noise?

Figure 3B-H: could the observation that the deletions of the distal enhancers reduce the noise associated with the mitotic lag indicate that long range E-P interactions are a contributory factor?

Figure 3E,J summarise results from an analysis of the mitotic lag wait times using a "staircase model". This is based on a previous paper but is not well elaborated here making it hard to follow how they arrive at the conclusions. As with other results in the paper, the $\text{sog}\Delta\text{prox}$ appears to be an outlier in its behaviour (with longer time intervals and single path).

The $\text{sog}\Delta\text{prox}$ is the most variable spatially (Figure 4) but differences are also seen with the sogWT . These are not discussed.

Promoter elements are investigated using synthetic reporters. Please can more information be provided. E.g. looking at the cited paper, it appears the sna enhancer used is the snaDist (i.e. the one remaining in $\text{sna}\Delta\text{prox}$).

The final modelling section is quite opaque and the conclusion that "model 2 successfully reproduces the steady state and produces a peak of FF" is not clear from the graphs shown (Figure 6H-K). And it's unclear how the modelling/data support the contribution from time-inhomogeneous bursting.

The discussion is quite lengthy. Many of the points covered are quite esoteric and could be condensed and some consideration given to the broader picture, for example the relevance of their findings in the context of fate decisions or otherwise.

Minor points:

Figure 5F -DPE crossed out in the scheme for twi (unmutated)

Figure 7A Red E1 should be E2 in the scheme.

Reviewer 3

Advance summary and potential significance to field

Using the MS2 live imaging and mathematical modeling, the authors analyzed how the regulatory code affects transcriptional noise in *Drosophila* embryos, in the context of redundant enhancers and promoter architecture. They examined sog and sna constructs, both wild type and the ones where one of the two redundant enhancers (proximal vs distal) was deleted. The Fano Factor was used as a measure of variability in this manuscript. It was shown that transcriptional noise peaks after mitosis and then decreases to steady-state levels later in the cycle. High noise after mitosis is mainly due to stochastic postmitotic reactivation timing, while parameters of transcriptional bursting mainly affect variability at steady-state. Redundant enhancers did not result in reduced noise, and enhancer strength did not necessarily correlate with the level of transcriptional noise. Promoter architecture, however, especially the TATA motif, significantly increased transcriptional noise. Lastly, mathematical models were used to show that both postmitotic lag time and time-inhomogeneous bursting are necessary to explain the observed noise patterns.

While live imaging has been used more commonly to characterize changes in transcriptional dynamics, analysis of the kinetics of variability has been largely missing. This manuscript works on this question. However, some of the statements are oversimplified and some conclusions are not well supported by the data. I'd like my comments to be resolved before reconsidering the publication of this work in Development.

Comments for the author

Major Comments

1. Lines 191-192 state "In the case of sog, the strongest enhancer leads to high noise, whereas for sna we observe the opposite." However, this interpretation appears oversimplified. For snail, the noise profiles are actually quite similar across all three constructs (Fig 2L), with the stronger sna^{WT} and sna^{ΔProx} showing higher levels at the beginning and the weaker sna^{ΔDist} exhibiting higher noise afterwards. The claimed "opposite" relationship is not clearly supported by the data.

2. Line 241-242 "These results suggest that mitotic lags are not obviously affected by the deletion of the snail proximal or distal enhancer, when assessed in the context of a reporter BAC transgene." As the authors pointed out, there is no significant difference in the lag time distribution from the modeling (Fig 3E). However, Figure 3C clearly shows that trajectory alignment does reduce noise levels for snail constructs, albeit more modestly than for sog. If mitotic lag distributions are truly unaffected by snail enhancer deletions (as the modeling suggests), then the reduction in noise upon temporal alignment needs explanation.

3. Throughout the manuscript, higher transcriptional activity generally leads to higher noise, but this relationship is neither explicitly stated nor rigorously tested. For example, when discussing TATA box effects (lines 367-368), the authors hypothesize that increased initiation rates contribute to noise, but the mechanistic connection is unclear. Is the argument simply that more frequent Pol II loading creates more opportunities for stochastic variation? If so, this should be explicitly stated. The authors should clarify whether they view the transcriptional activity-noise correlation as a trivial consequence of increased sampling or as a biologically meaningful finding.

4. Along the same line, when INR motifs don't affect noise but TATA boxes do, is this simply because TATA boxes have larger effects on transcriptional activity? Transcriptional activity is significantly higher when TATA motif is added, and this could cause higher pol ii loading rate and increase the noise. I don't see the INR transcription intensity/background data, so it is hard to conclude if that's the case.

5. Can the authors quantitatively separate contributions from different sources (transcriptional level, post-mitotic delay, bursting kinetics)?

6. Lines 367-368 state "We hypothesize that this difference in initiation rates may contribute to the increased transcriptional noise elicited by the TATA box." However, the mechanistic connection between higher initiation rates and increased noise is not clearly established. The authors' previous work (Pimmitt et al., 2021) demonstrated that TATA boxes increase Pol II initiation rates and transcriptional levels, but simply observing that both initiation rates and noise are elevated doesn't establish causality. Is the argument that higher initiation rates inherently lead to more noise, or is this simply a correlation with overall transcriptional activity? If noise simply correlates with transcriptional activity, then the TATA box findings may be less informative than suggested.

Minor Comments

Line 123 - need "used" in "We the MS2/MCP mRNA labeling system and.."

Fig 5F - the WT twi should have no "x" on DPE, as the twi enhancer contains DPE, no?

I didn't put everything, but there were other typos and errors in the manuscript and figures.

First revisionAuthor response to reviewers' comments

GENERAL

As you will see, the referees express considerable interest in your work, but have some significant criticisms and recommend a substantial revision of your manuscript before we can consider publication. Specifically, all three reviewers question whether the main finding, that redundant enhancers modulate transcriptional noise, is adequately supported by the data. The reviewers point out that the enhancers you studied exhibit distinct and sometimes opposing behaviours, with the clearest example of truly redundant enhancers (the *sna* series) showing minimal differences in noise levels. Furthermore, the reviewers raise the possibility that the observed changes in Fano Factor may reflect differences in burst strength that correlate with expression levels, rather than representing modulation of transcriptional noise. To address these concerns, the reviewers recommend further analysis of the data, including inferring the underlying bursting parameters across all experimental conditions.

We thank the referees for their careful evaluation of our manuscript and their constructive comments. We agree that, with the genetic material currently available, we cannot draw a definitive conclusion about how redundant enhancers affect noise. Although this question was part of our original motivation, our primary objective was not to resolve it, but rather to investigate more broadly how cis-regulatory sequences shape transcriptional noise in space and time. We also note that neither the title nor the abstract makes a definitive claim regarding redundant enhancers.

We fully agree with the referees that (1) only a limited number of enhancer pairs were analyzed and (2) the *sog* enhancer pair is not strictly 100% redundant, which is why we initially described them as “seemingly” redundant. In the revised manuscript, we have therefore revised our conclusions about enhancer contributions to noise to avoid any ambiguity regarding redundant enhancers.

- We explicitly note this limitation in the Results (line 199), the Discussion (line 495), and the Limitations of the Study section (line 558).
- We systematically replaced “redundant enhancers” with “shadow enhancers,” a more agnostic term regarding functional overlap. We also prefer this terminology because it is used in the original publications describing the *sog*-MS2 alleles (published in *Development*) and the *sna*-MS2 alleles (DOI: 10.7554/eLife.07956) used in this study.
- We modified the summary figure (Figure 7) to remove the initial panel regarding redundant enhancers.

Color code for the Point-by-point answer:

Reviewer comments are in black.

Our answers are in blue.

Quotes from the revised text are in green.

Reviewer 1: SUMMARY OF THE ADVANCE MADE IN THIS PAPER AND ITS POTENTIAL SIGNIFICANCE TO THE FIELD

Maillard et al provide detailed quantifications of transcriptional dynamics (MS2-MCP system) for two developmentally important genes (*sna* and *sog*) across a series of CRE deletions. The paper's strengths are the careful live imaging, and detailed temporal analysis. In principle, comparisons of the contributions of “redundant”/“shadow” enhancers to transcription and associated noise is important to understand how multiple regulatory inputs are integrated to achieve cell fate decisions during development, and to do so robustly. Efforts to combine detailed measurement with quantitative modelling is key to solve this problem, and is something that the authors are helping contribute towards.

SUGGESTIONS TO AUTHORS

Overall, the manuscript suggests that redundant enhancers modulate transcriptional noise, but this conclusion is not fully supported by the data, as the enhancers studied show differing behaviours

and the clearest redundant pair shows minimal noise differences. Additionally, observed changes in Fano Factor may largely reflect differences in burst strength linked to expression level, rather than direct modulation of transcriptional noise. Before publication, the following major points should be addressed:

We thank the referee for their careful review of our work and for recognizing both the fundamental question we address and our efforts to combine detailed measurements with quantitative modeling.

1.1 Misinterpretation of Fano Factor (FF) and Noise

Fano Factor appears to scale with mean expression, which is likely due to increased burst strength. This is distinct from a true modulation of transcriptional noise.

The authors claim opposing behaviour in *sog* vs *sna*, but these differences if any are not clear at all, given the above: in both cases, higher-expressing constructs show FF peaks, as expected if burst strength increases.

The referee suggests that the FF may scale with the mean expression length and that our measurements do not reflect a direct modulation of transcriptional noise. While mean expression levels can in some cases correlate with FF, it is not always the case and there is no causal relationship between these two metrics.

To directly assess whether the mean and the Fano factor were correlated in our experimental data, we calculated the mean and the Fano factor (within the stable, long-term region) and log-log plotted them against each other for all genotypes (**Figure S5**). If a linear (or power law) correlation were present, one would expect to observe a straight line; however, no such trend is evident. Instead, the mean can vary over several decades without a corresponding significant change in the Fano factor.

We would like to emphasize that our choice of Fano is the result of careful and thorough consideration. We have chosen a unitless variable (number of events in a time interval) because we know that the contrary (choosing the Fano factor of the intensity) would make Fano dependent on arbitrary units. For the Fano chosen we have a mathematical theory that we briefly exposed in the Methods.

From a theoretical point of view, it also appears that FF and the mean expression are independent metrics. This is now fully discussed in the Methods (in relation with Eqs.8-11) and in the Supplementary Text.

We hope that, with these clarifications and the direct evidence demonstrating the absence of a clear correlation between mean and the Fano factor (**Figure S5**), the referee will reconsider their conclusion regarding our alleged “*misinterpretation of the Fano factor and noise.*” In order to clarify this key point in the manuscript, we added a few explanations (line 189) and provided this plot as a new supplementary **Figure S5**. We also rephrased the introductory explanation regarding FF and bursting (line 127).

Figure S5: Comparison of mean expression and Fano Factor for all tested genotypes and regions.

1.2. Misuse of the Term "Redundant Enhancers"

The enhancers examined show distinct and sometimes opposing regulatory behaviour, with one (*sog* proximal) potentially acting as a repressor, undermining the claim of redundancy.

The *sna* enhancer series—the only true redundant pair—shows no difference in noise, which contradicts the main claim that redundant enhancers modulate transcriptional noise.

This work is not focused on how redundant enhancers affect noise, as this is neither the message conveyed by the title nor by the abstract. We apologize if the initial version of the manuscript gave this impression. We revised the text to place less emphasis on the specific question of redundant enhancers and noise. Indeed, we fully acknowledge that we have analyzed only two enhancer pairs and that they are not strictly 100% redundant. We explicitly describe their distinct behaviors (for *sog*, see line 147, and line 152). In the revised manuscript we now refer to the more agnostic term ‘shadow enhancers’, following the terminology introduced in the original publication from the Rushlow lab that generated the *sog*-MS2 CRISPR alleles (Whitney *et al.*, *Development*) and that of the Levine lab for *sna*-MS2 alleles (Bothma *et al.*, *elife* 2015).

We revised the abstract, deleted the sentence regarding redundant enhancers and replaced ‘characterized the contribution of seemingly redundant enhancers to noise’ by ‘assessed the contribution...’.

1.3. Lack of Inferred Bursting Parameters

The authors should systematically infer burst frequency, strength, initiation intensity, and duration from their data. They only attempt this for one promoter (TATA box example), even though this analysis is critical to properly interpret all experiments.

Without these parameters, it's unclear whether the Fano Factor differences actually reflect modulation in transcriptional noise, rather than some burst correlates of mean expression.

While we agree with the reviewer that inferring bursting parameters for all genotypes would indeed have been informative, we respectfully disagree that this information is essential for the proper interpretation of our data.

First, as discussed in point 1.1, the Fano factor does not correlate with mean expression in our dataset and we therefore interpret it as reflecting a genuine modulation of transcriptional noise. In

our framework, the Fano factor serves as a proxy for bursting: it equals one in the absence of bursting (i.e., under Poisson noise) and increases with the degree of bursting. While we would have preferred to include additional bursting parameters for a more detailed characterization, this was not feasible for most genotypes, with the exception of *twist*.

Second, over the past decade, our research has consistently focused on inferring quantitative bursting parameters, and we have developed rigorous methods to infer promoter models with two, three, four, or more states (Douaihy *et al.*, 2023 NAR, PMID: 37522372; Radulescu *et al.*, 2025 Bull. Math. Biol., PMID: 39625575). The first step of these methods is a deconvolution of the signal in a series of transcription initiation events, which can be applied to any signal, including time-inhomogeneous ones. However, the second step, estimating state transition rates, requires stationarity, that is, the assumption that transcriptional kinetics do not vary over time. In prior work, we have systematically verified this stationarity assumption before applying the second step of our *BurstDeconv* method (Pimmitt *et al.* 2025a, b; PMID: 40018801; PMID: 40890123). For example, to infer bursting parameters for endogenous *snail* transcription, we first identified time windows during which the RNA production rate, ($pON \cdot K_{ini}$), was stationary using a Bayesian Change Point Detection approach (Pimmitt *et al.* 2025b, Nature Communications). We did the same here when we determined the rate parameters for *twi* in a window during which $pON \cdot K_{ini}$ was constant, but were not able to do so for all the genotypes.

Accounting for time-dependent dynamics would require estimating additional parameters. While this is feasible for some summary parameters such as $pON \cdot K_{ini}$ (which is obtained directly from deconvolution and has also been used by others, including the Gregor lab (Chen *et al.*, 2025 NSMB), it cannot be extended to all parameters without risking overfitting.

Figure for Reviewers: RNA production rate ($pON \cdot K_{ini}$) for *sog*, *sna* and *twi* MS2 allele series in first 30 minutes of nc14. (A-D) RNA production rate for *sog*-MS2 in (A) dorsal, (B) dorso-lateral, (C) ventro-lateral and (D) ventral regions. (E) RNA production rate for *sna*-MS2 reporters. (F) RNA production rate for *twi*-MS2 reporters.

Statistics: dorsal region: *sog*^{WT} N= 2 embryos, n= 126 nuclei, *sog*^{ΔProx} N= 4 embryos, n =165 nuclei. dorso-lateral region: *sog*^{WT} N= 4 embryos, n= 286 nuclei; *sog*^{ΔDist} N= 3 embryos, n= 84 nuclei; *sog*^{ΔProx} N= 5 embryos, n =407 nuclei. Ventro-lateral region: *sog*^{WT} N= 5 embryos, n= 306 nuclei; *sog*^{ΔDist} N= 5 embryos, n= 242 nuclei; *sog*^{ΔProx} N= 4 embryos, n= 333 nuclei. Ventral region: *sog*^{WT} N= 3 embryos, n= 140 nuclei; *sog*^{ΔDist} N= 5 embryos, n= 220 nuclei; *sog*^{ΔProx} N= 3 embryos, n= 121 nuclei; *sna*^{WT} N= 5 embryos, n= 404 nuclei; *sna*^{ΔDist} N= 4 embryos, n= 248 nuclei; *sna*^{ΔProx} N= 5 embryos, n= 405 nuclei; *twi* N= 3 embryos, n= 485 nuclei; *twi-DPE* N= 3 embryos, n= 357 nuclei; *twi-DPE+TATA* N= 3 embryos, n= 451 nuclei.

Unfortunately, for most genotypes concerned by this manuscript, transcription dynamics are not stationary. We now show this information as a figure for the reviewer, depicting the evolution of the RNA production rate, quantified as $pON \cdot K_{ini}$, across *nc14* for all genotypes. The plots show that, with the exception of a few genotypes (e.g. *sog* WT, Fig. for Reviewers), transcription rates are not stationary.

Using the aforementioned Bayesian Change Point Detection approach, we tried to isolate windows where pON would be stable, however, these intervals were not long enough to enable reliable inference of promoter state dynamics and bursting parameters. Although it would, in principle, be possible to infer a model with time-dependent parameters, this lies beyond the scope of the present work, and the risk of overfitting would remain. Note that for inferring *twist* promoter dynamics, we specified in the text line 348 ‘*The mean transcription rates of the promoter evolve during nc14, progressively decreasing before reaching a plateau around 20 minutes into nc14 (Fig. S4A). We therefore focused our attention on the 20-30 min period, during which transcription had reached steady state (Fig. S4A).*’

For all these reasons, we prefer to use the Fano Factor as a single, comprehensive summary of bursting.

1.4. Missing or Incomplete Data Presentation

Mean expression traces are missing or relegated to the supplement, making it challenging/impossible to interpret the FF plots.

All aligned analyses need to show the corresponding mean expression and aligned time traces. Likewise for the TATA box analysis

We have now included the mean expression plots in the main manuscript. For *sog* data, mean expression is shown in Figure 2C and Figure 4 E, I, M. *snail-BAC* mean expression is shown in Figure 2H. *twist* transgene expression is shown in Figure 5H. For the *snail-promoter* transgenes, the data are published and the plots can be found in Pimmitt *et al.*, 2021 *Nature Communications*, PMID:34301936.

1.5. Overstated and Unsupported Conclusions

- The main conclusion—that noise differs between redundant enhancers—is not supported: those that differ appear not to be redundant; those that are redundant appear show no difference. We nuanced our conclusions as we agree that we only surveyed 2 model loci, that are in fact not representative of a true situation of enhancer redundancy (if this situation ever exists). For example, we changed the title of section line 135 to replace ‘redundant enhancers’ with ‘*Enhancers with overlapping functions differentially affect transcriptional noise*’.

While we agree that *sog* enhancers are far from being redundant, they do exhibit overlapping activities as carefully described by the Rushlow lab, in a manuscript published in *Development* (Whitney *et al.*, 2022 PMID:36264246). We clearly stated the similarities and distinct properties of these enhancers, when introducing them line 147.

We reformulated the abstract to give less emphasis on the conclusion regarding redundant enhancers and noise. We also modified the summary Figure 7 to remove the initial panel on redundant enhancers. We also clearly stated this limitation in the ‘limitation of the study section’

line 558 with the sentence: ‘Given the small number of enhancers examined in this study and their partially overlapping functions, we cannot rigorously investigate the effect of shadow enhancers on transcriptional noise’.

Additionally, some further points should be responded to:

The paper presents two possible interpretations of transcriptional noise: (a) intrinsic, internal fluctuations at a single locus over time, and (b) inter-nuclear or population-level variability measured across nuclei within time windows. The apparent reliance on inter-nuclear variability could be inconsistent with models based on intrinsic noise. The authors should clearly define which type of noise they are measuring, what assumptions they’re making therein, justify its use in the context of their modelling, and, if necessary, re-analyse their data to distinguish between intrinsic and population-level noise sources.

We thank the referee for this question. To address this point, we have imaged embryos expressing 2 copies of MS2 reporter transgenes, monitored with an MCP-GFP detector (2 spot movies but with one color). We have then implemented 2 spot analysis in our image analysis pipeline to independently track each individual transcription site, in single nuclei. These two reporter experiments allow us to estimate the relative contributions of intrinsic and extrinsic noise. To quantify the data, we used the method of Elowitz et al. extended from static measurements to time series in Topno et al., 2025 (this methodology is detailed in DOI: 10.1101/2025.11.26.690455, currently under revision at *Science Advances*) to quantify both intrinsic and extrinsic noise. Our analysis indicates that in this context, transcriptional noise is primarily intrinsic. These new results are presented as a new supplementary figure (Figure S6), detailed in the methods line 1050 and discussed in the manuscript line 408-422.

Figure S6: A two-reporter construct is used to separate intrinsic from extrinsic noise. (A) Sites are considered *paired* when they are in the same nucleus and *unpaired* when they are randomly drawn from different nuclei within a defined spatial domain. (B) Schema of *sna* and *sna+INR* reporters. (C-D) Decomposition of the noise variance is performed for *sna* (C) and *sna + INR* (D). The total

variance is the sum of intrinsic and extrinsic noise and should be similar whether it is computed using paired or unpaired sites. The paired variance corresponds to extrinsic noise and is the sum of two components: intracellular and extracellular. The unpaired variance reflects only the extracellular component of extrinsic noise. The intracellular extrinsic component arises from fluctuations that are shared by two alleles within the same nucleus but vary independently between nuclei, such as transcription factor concentration, whereas the extracellular extrinsic component arises from fluctuations that are common to all alleles, regardless of whether they are in the same nucleus or not.

Another argument supporting that transcriptional noise in our biological context is primarily intrinsic comes from our modeling. In Figure 6, we did not fit a model to the data; rather, we used bursting parameters derived from the distribution of waiting times in single-cell transcription data. By definition, these parameters describe intrinsic noise. Assuming only intrinsic noise, we were able to reproduce the Fano factor curve. If extrinsic noise were present in addition to intrinsic noise, the Fano values at large times would have been underestimated by our theoretical model. The fact that they are not suggests that extrinsic noise is small. We added a sentence to mention this observation line 411.

The manuscript could benefit from clearer presentation, both in the text and figures. Axis scales are missing in places, and formatting at times is inconsistent, and there are plenty of typos.

We apologize for the few errors that escaped our attention and have done our best to capture them all.

Line 159-160: Figure 2D plots the absolute intensity of transcripts. It is therefore immediately obvious to distinguish which allele has "more variable transcription". Reference to the "y-axis" here doesn't help much.

We are uncertain here about the suggestion of the reviewer as we don't show absolute intensity but intensity/background. The intensities are shown, as well as the SEM across embryos. We believe that this graph is OK.

In Figure 2C-D, it appears that Δ Prox shows a higher peak of expression than the WT. Can the authors comment somewhere whether this proximal element is thus functioning as a (partial) silencer? Whether these sequences have positive or negative influences on transcription alters the interpretation of noise regulation.

We prefer not to discuss this point in depth, as our aim is not to characterize the function of each cis-regulatory element (as enhancer/silencer). This was extensively performed in previous studies by the Rushlow and Stathopoulos labs. The relative contribution of these sequences to transcription is already briefly described in our manuscript: line 145 and line 276.

Line 185: "the control" should be described as sna^{WT} . Done.

Lines 195-201: This paragraph lacks any references to data. It is unclear whether this is new observations, re-interpretations of old ones (which would be quite a stretch), or speculation.

We thank the reviewer for this comment. Indeed, this paragraph was an interpretation and speculation from the data presented in Figure 2. In the revised manuscript, we removed this paragraph, as it is discussed in the discussion section, with more nuance.

A more detailed description of how the Erlang (γ) model is fit is warranted. For presenting results, consider presenting something more than a schematic (Fig. 3J), for example providing evidence for model fit, and some sense of inter-embryo variability in measurements. It appears that using Erlang model makes sense only if considering temporal noise of a single cell, or a reasonable proxy thereof (see above).

The authors should discuss their work in relation to "Alamos, S., Reimer, A., Westrum, C., Turner, M. A., Talledo, P., Zhao, J., Luu, E., & Garcia, H. G. (2023). Minimal synthetic enhancers reveal control of the probability of transcriptional engagement and its timing by a morphogen gradient. *Cell Systems*, 14(3), 220-236.e3. <https://doi.org/10.1016/j.cels.2022.12.008>", which also considers Dorsal response, transcriptional induction, multi-step transcriptional initiation and mitotic entry contributions.

We thank the referee for this reference and we now refer to it in our text (line 225). We originally introduced the use of mixed Erlang distributions to model post-mitotic reactivation in 2018 (Dufourt et al., 2018, *Nature Communications*; PMID: 30518940), i.e., prior to Alamos et al. (2023). This approach is motivated by a mechanistic view in which, following mitotic pausing, transcription resumes via an irreversible sequence of steps (the "staircase model"; Dufourt et al., 2018; Bellec et al., 2022, *Nature Communications*; PMID: 35246556), with step timescales modulated by transcription factors such as Zelda.

Alamos et al. (2023) propose a closely related idea: the promoter must traverse several kinetic steps before transcription, and changes in TF binding affinity or TF concentration alter the corresponding rates, thereby affecting both the engagement probability and the onset-time distribution. We understand that the referee is suggesting consistency with this concept—namely, that if the transcription factor concentration varies over time, then the transition timescales should also be time-dependent. Our framework covers this possibility because we can consider inhomogeneous, rather than homogeneous times for the different steps. However, the concentrations of the relevant regulators change only later in *nc14*, well after mitosis, which makes the homogeneous time approximation reasonable for modeling mitotic reactivation. Furthermore, trying to infer a time-inhomogeneous model produced overfitting (the number of parameters to fit becomes too large) without increasing the quality of the fit.

As suggested by the referee we now provide further evidence for the model fit (**Figure S7**). The method has been already published in Dufourt et al., 2018 and we therefore believe that it does not need an extensive presentation. It is briefly summarized in the methods. The mixed Erlang distributions depend on parameters M, p_1, p_2, \dots, p_M , and T , representing the integer number of steps (shape parameter), the probabilities to use one, two, or M steps to reactivate after mitosis and the mean time for each step. We compare models with increasing M . M is selected according to a parsimony principle: it is the smallest M for which the predicted survival function of the activation time is within the confidence interval of the empirical survival function (computed using Greenwood's formula for variance). Together with the objective function value, the confidence interval test is a way to test the quality of the fit. This step is important and illustrated by the figure below. As can be seen $M=2$ is not a good model but $M=3$ is.

Figure S7: A mixed Erlang model fitting of the distribution of postmitotic reactivation times. (A-C) Post-mitotic reactivation time fitting for sna^{WT} (A), $sna^{\Delta Dist}$ (B) and $sna^{\Delta Prox}$ (C) for $M=2$ and $M=3$ steps. (D-F) Post-mitotic reactivation time fitting for sog^{WT} (D), $sog^{\Delta Dist}$ (E) and $sog^{\Delta Prox}$ (F) for $M=2$ and $M=3$ steps. Empirical estimates of the survival function of these intervals using the Kaplan-Meier method are shown (blue) with 95% confidence intervals (orange) computed with Greenwood's formula and predicted survival function (red). A good fit occurs when the predicted survival function is within the bounds of the confidence interval. A good fit can be obtained by increasing the number of parameters in the mixed model, i.e. the maximal number of steps M .

Methods appear to be missing for how "Cumulative activation percentage (mean \pm s.e.m. over embryos) by genotype" is calculated. Does "activation" imply intensity above some low fixed threshold, or is the threshold variable, or is it something else. Is this the proposal of "the one-fifth threshold" (Line 888)? If so, can the authors appreciate how altered time-scale of activation in various mutants could be undermined/mis-interpreted due to different maximum values?

We apologize to the reviewer for the gap in clarity. With respect to the cumulative activation percentage, activation is marked at the first detectable transcription event in the nucleus. This threshold is dependent on the specific imaging settings, which are kept constant for all reporters within a particular gene series so as to remain comparable. This has been added to the methods section line 925. The '*one-fifth threshold*' is a simplified way to identify the mitotic lag. Although the distribution of the mitotic lag and the cumulative activation curve are related, these are computed using different methods.

"This suggests that the necessary and sufficient condition for observing a peak in the Fano factor is to have a postmitotic lag, followed by time-inhomogeneous bursting." This extremely strong claim is not fully warranted. The modelling suggests its sufficient, but there are likely many possible mechanisms (Lines 405-406)

We agree with the referee and have nuanced our wording. It is a sufficient condition, not obviously necessary. This sentence is now replaced line 406 by *'This suggests that a postmitotic lag, followed by time-inhomogeneous bursting, is a minimal explanation of a peak in the Fano factor.'*

In Table S3, the confidence intervals are remarkably tight (identical to 4-5 significant figures). Are these those that come out of the BurstDECONV software. It would be good to have a brief description of how to interpret these. Did the authors try fitting these separately by embryo, or via bootstrapping, to get a sense of the real variance.

This is an "uncertainty" interval and was not obtained by bootstrapping, but from the distribution of suboptimal parameters for which the objective function takes values between optimum and optimum multiplied by an overflow larger than one. This method (part of BurstDeconv) tests global sensitivity of the objective function: a very sensitive function confines the parameters to a narrow interval, whereas a less sensitive function allows large variations (uncertainties). We added these explanations to Methods.

Minor comments:

stray comma in Line 76. **Ok**

Lines 91-92: Appreciate the more general description of "redundant enhancers", but readers may benefit from their historical and perhaps better known descriptor "shadow enhancers".

We agree with this suggestion and have now replaced 'redundant enhancer' by the more agnostic and historical term 'shadow enhancers'.

Line 96: "fluctuations on..." should probably be "fluctuations in...". **done**

Line 133: stray comma in "Transcriptional bursting, results" **done**

Line 142: "Dorsal and are essential ..." typo. **In the context of the full sentence, this is not a typo.. The text read as 'Both gene products are activated by the morphogen Dorsal and are essential for patterning the dorso-ventral axis (Leptin, 142 1991).'**

Figure 1C bottom. The "n" in "tn" should be subscripted for consistency. **Done**

Line 178: Is there need to spell out "sna delta distal/proximal"? Especially given this is not provided for the sog alleles. **We believe this improves the clarity of the text.**

Line 210: sub-clause "particularly the enhancer" needs re-writing for improved clarity.

We now deleted the part 'particularly the enhancer' (line 204), as it is not required.

Line 213: Missing space "bookmarking(Bellec et al., 2022 ...)". **Done**

X-ticks are mixing in Fig 2D and 2I. Single trace panels have now been moved to the supplements: Previous Fig2D/I are now Figure S1 C, F and depicted in much larger views to facilitate their reading. Although small, x-ticks are present in these panels (15 and 30 minutes) and we feel adding more would not meaningfully contribute to the clarity of the data.

Figure 3: it would help to have the sog on the left and sna on the right for consistency with Figure 2.

We agreed and initially wanted to present this order; however this does not follow our text. We therefore prefer to keep this order in the figure.

It should be cited somewhere that Fig3B and G are repeated from Figure 2.

We added this note in the Figure legend of Figure 3. Line 621: 'Note that panel B/G are repeats of Figure 2 panels E/J'.

Given repeated data, is there a reason to show different y-axis extents for Fig 3B and Fig 2L.

Yes, the y-axis was changed to facilitate the comparison between aligned/non aligned FF, shown in Figure 3 C/H, as it's the goal of this Figure.

Raw intensities in the various reporters analysed in Figure 5 should be reported somewhere, so readers can distinguish intensity variations from noise.

As suggested by reviewers, for all genotypes imaged in this manuscript, each FF/noise panel is now presented with the corresponding intensity panel. The intensity panels for the *snf4Enhancer-promoter-MS2* transgenes presented in Figure 5B-E are published in Pimmett *et al.*, 2021 in Figure 5C as sample traces. Please note that we now provide a figure showing the absence of a clear correlation between intensities and noise (new Figure S5).

Heat map in Figure 6 should use stronger colours (e.g. Greyscale) so that the "white intervals" referred to in the text are visible.

Figure 6B,C have been regenerated so as to highlight more visually the non-productive intervals. We note that we are constrained as to the dynamic range of the gradient with respect to the number of initiating PolII so as to maintain comparability between the genotypes.

Models 1 and 2 in Figure 6 should have corresponding schematics so readers can understand their differences clearly.

Following Rev1 suggestion, we added schematics to Figure 6.

Methods: missing space in "The mappingsfrom multiexponential" (line 935).

Thanks for noting this, corrected (now line 972).

Missing close bracket in Line 830.

Done.

Reviewer 2:

SUMMARY OF THE ADVANCE MADE IN THIS PAPER AND ITS POTENTIAL SIGNIFICANCE TO THE FIELD

The paper sets out to investigate sources of transcriptional noise and how they might contribute to developmental fate decisions. They utilize the elegant live-imaging approaches for which the Lagha lab is well-known and combine that with sophisticated analysis and modeling to tease out the underlying principles.

Strengths include a thoughtful description of transcription noise, decoupling changes in intensity from noise and inter-nuclear variability. They also apply a consistent parameter of noise (F-Factor) throughout, using it to measure noise across scales of temporal, cell-cycle and spatial based variabilities. The results highlight that the heterogeneity in the reinitiation of transcription following exit from mitosis is one major source of transcriptional noise. Notably, the coordinated cell divisions in the *Drosophila* embryo provided a good vehicle to detect this feature, which would be hard to distinguish in heterogeneous populations. They discuss the factors that might influence this aspect of noise ("mitotic lag") but they don't consider the potential impact for cell fate decisions- might this contribute to a period when cells are more plastic after mitosis for example?

The authors also investigate contributions from promoter elements, turning to defined synthetic reporters in place of the engineered gene loci. Their results that promoter elements impact differently on the levels of transcriptional noise and the final models suggest that the promoter kinetics are one factor influencing noise.

The results are interesting and the analysis informative, especially with respect to the mitotic lag. There are some concerns about the mixture of genetic tools that have been used and whether these might confound the interpretations. Nevertheless, the results bring valuable insights. For those to be accessible to a wide readership, the authors are recommended to clarify the text and the arguments as in the suggestions below.

We thank the reviewer for his/her in depth reading of our manuscript. We are happy to see that this reviewer appreciated our efforts to 'decouple intensity from noise' and considers that our results bring 'valuable insights'.

SUGGESTIONS TO AUTHORS

The results are quite confusing for a reader who is not deeply familiar with these types of data and analysis. In some places very little explanations are given, and the reader has to refer to previous papers to understand the methods. They are also quite selective in the results that they emphasize, and it is not always clear why.

The *sog* Δ *prox* has the most distinctive behaviour in all of the assays. This variant behaves very differently from all the others tested (but more similar to the synthetic reporters?) Is there a concern that this is behaving anomalously - Could the hyperactivity of the *sog* Δ *prox* be a consequence of the deletion being within the intron, impacting for e.g. on RNA stability?

The reviewer is correct in the fact that the *sog* Δ *prox* genotype is the most distinctive. The enhancer has not been deleted but was replaced by a synthetic sequence of equivalent size. We do not believe that this editing affects splicing or RNA stability, as suggested by the phenotypic analysis performed in Whitney et al., 2022. Indeed, if the enhancer replacement was affecting splicing and/or mRNA stability, one would not expect to obtain only 6% of lethality with this genotype.

The changes (decrease in intensity and F-factor) in both *sog* Δ *Dist* and *sna* Δ *Dist* (Fig 2C,J,F,L) are not discussed or elaborated on. In the initial experiments they state that *sog* Δ *prox* "shows more variability" than *sog*WT or *sog* Δ *dist* referring to Figure 2D. The graphs there have very different scales, but as presented, the *sog* Δ *prox* seems to have the least variability (highest intensity but fairly smooth traces). Better presentation of the data is needed to clarify this point. (these graphs are very pixelated and vary in scales, presentation could be improved).

We now present enlarged views of single nuclei traces, with the same scale, in Figure S1 C/F to better grasp the variability in transcriptional activities between neighboring nuclei. We apologize for the poor quality of the initial panels and hope that these panels now better reflect the variability.

One of their initial conclusions is that the intrinsic noise is not related to enhancer strength. But the effects from deleting *sna* enhancers on levels of expression (the measure of strength) are relatively minor unlike those with the *sog* where there is a large-scale change.

Indeed, our conclusion is that intrinsic noise is not related to enhancer strength. We now show this result more systematically, for all genotypes with a graph documenting the absence of correlation between FF and intensities (Figure S5). See also response to Reviewer1, point 1.1.

Don't the results suggest that normally the *sog*-*prox* enhancer helps to suppress noise? Figure 3B-H: could the observation that the deletions of the distal enhancers reduce the noise associated with the mitotic lag indicate that long range E-P interactions are a contributory factor?

We believe that the high noise observed in the *delta*-*prox* is not primarily due to the absence of this intronic enhancer but rather to the fact that in this genotype, transcription solely relies on a distal enhancer, located 20kb away that needs to come in proximity to the promoter. This additional search process may represent an extra regulatory step for transcription to occur and is probably responsible for the observed high noise.

We note however that since the revised manuscript now puts less emphasis on the question of shadow enhancers and noise, we prefer not to discuss in depth why *sog*-proximal may act as a noise

suppressor. We now added a sentence in the 'limitation of the study section' line 558 to reflect this point. Please also see response to ref1, point 1.5.

Figure 3E,J summarise results from an analysis of the mitotic lag wait times using a "staircase model". This is based on a previous paper but is not well elaborated here making it hard to follow how they arrive at the conclusions. As with other results in the paper, the $sog\Delta prox$ appears to be an outlier in its behaviour (with longer time intervals and single path). The $sog\Delta prox$ is the most variable spatially (Figure 4) but differences are also seen with the $sogWT$. These are not discussed.

We have developed the part of Methods describing the implementation of this method. The referee correctly identified $sog\Delta prox$ as having different activation pathways. Although we do not have a model for this, we believe that changes of the enhancer organization may have a strong influence on the paths to reactivation after mitosis. Deleting the proximal enhancer could force the system to rely on alternative, slower pathways to reactivation, revealing hidden intermediate states. We opted for not discussing in depth this observation as we prefer to give emphasis on the effect of the mitotic lag time on transcriptional noise, a common feature observed for all genotypes. As pointed by all referees, our limited number of enhancers does not allow us to conclude on the effect of enhancers on transcriptional noise. This is now clearly mentioned in the discussion line 495 and limitations of the study line 558.

Promoter elements are investigated using synthetic reporters. Please can more information be provided. E.g. looking at the cited paper, it appears the *sna* enhancer used is the *snaDist* (i.e. the one remaining in $sna\Delta prox$).

We added to the first section of the methods that its *distalCore*.

The final modelling section is quite opaque and the conclusion that "model 2 successfully reproduces the steady state and produces a peak of FF" is **not clear** from the graphs shown (Figure 6H-K). And it's unclear how the modelling/data support the contribution from time-inhomogeneous bursting.

We have added quantitative details to the modeling section, including a schematic in Figure 6 for clarity. Figure 6 compares the two models, showing that the peak is generated by Model 2 but not Model 1, leading us to reject Model 1 and retain Model 2. Furthermore, the parameters of the retained Model 2 were not fitted directly to the time dependence of the Fano factor (and therefore not to the peak value); instead, the kinetic parameters were determined from segments of the live transcription signal, assuming constant rates within each segment. This strengthens the validity of the model, which relates the kinetic parameters and their approximate time dependence to the Fano factor; even if the agreement with the Fano curve is not perfect, it remains within the confidence interval.

The discussion is quite lengthy. Many of the points covered are quite esoteric and could be **condensed** and some consideration given to the broader picture, for example the relevance of their findings in the context of fate decisions or otherwise.

As suggested by the referee, we now reduced the discussion (line 478-480 deleted, line 494-495 deleted, paragraph line 502-514 deleted, line 541 deleted and deletions line 551).

Minor points:

Figure 5F -DPE crossed out in the scheme for *twi* (unmutated).

Thanks, now fixed.

Figure 7A Red E1 should be E2 in the scheme.

Yes you are correct, now fixed.

Reviewer 3: SUMMARY OF THE ADVANCE MADE IN THIS PAPER AND ITS POTENTIAL SIGNIFICANCE TO THE FIELD

Using the MS2 live imaging and mathematical modeling, the authors analyzed how the regulatory code affects transcriptional noise in *Drosophila* embryos, in the context of redundant enhancers and promoter architecture. They examined *sog* and *snail* constructs, both wild type and the ones where one of the two redundant enhancers (proximal vs distal) was deleted. The Fano Factor was used as a measure of variability in this manuscript. It was shown that transcriptional noise peaks after mitosis and then decreases to steady-state levels later in the cycle. High noise after mitosis is mainly due to stochastic postmitotic reactivation timing, while parameters of transcriptional bursting mainly affect variability at steady-state. Redundant enhancers did not result in reduced noise, and enhancer strength did not necessarily correlate with the level of transcriptional noise. Promoter architecture, however, especially the TATA motif, significantly increased transcriptional noise. Lastly, mathematical models were used to show that both postmitotic lag time and time-inhomogeneous bursting are necessary to explain the observed noise patterns.

While live imaging has been used more commonly to characterize changes in transcriptional dynamics, analysis of the kinetics of variability has been largely missing. This manuscript works on this question. However, some of the statements are oversimplified and some conclusions are not well supported by the data. I'd like my comments to be resolved before reconsidering the publication of this work in Development.

SUGGESTIONS TO AUTHORS

Major Comments

1. Lines 191-192 state "In the case of *sog*, the strongest enhancer leads to high noise, whereas for *snail* we observe the opposite." However, this interpretation appears oversimplified. For *snail*, the noise profiles are actually quite similar across all three constructs (Fig 2L), with the stronger *snail*^{WT} and *snail*^{ΔProx} showing higher levels at the beginning and the weaker *snail*^{ΔDist} exhibiting higher noise afterwards. The claimed "opposite" relationship is not clearly supported by the data.

We agree with the referee and we have reformulated this paragraph. We have now deleted this sentence (line 188) and added a reference to the new Figure S5, showing the absence of obvious correlation between enhancer strength and noise. Please also see the answer to Ref1 point 1.1.

2. Line 241-242 "These results suggest that mitotic lags are not obviously affected by the deletion of the *snail* proximal or distal enhancer, when assessed in the context of a reporter BAC transgene." As the authors pointed out, there is no significant difference in the lag time distribution from the modeling (Fig 3E). However, Figure 3C clearly shows that trajectory alignment does reduce noise levels for *snail* constructs, albeit more modestly than for *sog*. If mitotic lag distributions are truly unaffected by *snail* enhancer deletions (as the modeling suggests), then the reduction in noise upon temporal alignment needs explanation.

At steady state (i.e. at long times), the Fano Factor differs substantially across the various *snail* deletions, with *snail*^{ΔDist} exhibiting larger values than the others. In the Methods, we provide analytic expressions that relate the parameters of the survival function and thus the kinetic rate parameters of steady-state bursting (Eqs. 2, 6, 7), to the steady-state Fano. These formulas make explicit how the steady state Fano depends on the bursting parameters. Thus, even if the mitotic lag parameters (which describe a distinct process and are not part of the bursting dynamics) are identical, differences in bursting parameters alone can account for the observed differences in Fano. We therefore conclude that, for *snail*, the observed differences in Fano factor result from differences in the bursting parameters.

3. Throughout the manuscript, higher transcriptional activity generally leads to higher noise, but this relationship is neither explicitly stated nor rigorously tested. For example, when discussing TATA box effects (lines 367-368), the authors hypothesize that increased initiation rates contribute to noise, but the mechanistic connection is unclear. Is the argument simply that more frequent Pol II loading creates more opportunities for stochastic variation? If so, this should be explicitly stated.

The authors should clarify whether they view the transcriptional activity-noise correlation as a trivial consequence of increased sampling or as a biologically meaningful finding.

Please refer to our response to reviewer 1, comment 1.1 and accompanying new figure, Figure S5.

4. Along the same line, when INR motifs don't affect noise but TATA boxes do, is this simply because TATA boxes have larger effects on transcriptional activity? Transcriptional activity is significantly higher when TATA motif is added, and this could cause higher pol ii loading rate and increase the noise. I don't see the INR transcription intensity/background data, so it is hard to conclude if that's the case.

Please refer to our response to reviewer 1, comment 1.1 and accompanying new figure, Figure S5.

5. Can the authors quantitatively separate contributions from different sources (transcriptional level, post-mitotic delay, bursting kinetics)?

At steady state the only source of heterogeneity is constant bursting kinetics. The transient peak in the Fano factor is given by both post-mitotic lag and time-inhomogeneous bursting kinetics. We can indeed separate the effect of constant bursting kinetics from the rest. This is already visible by comparing Figure 3B-G and Figure 3C-H. This shows the loss of the post-mitotic lag time can contribute to 50%-75% of the Fano peak. We added this sentence to the text line 226.

“The contribution of various sources is variable in time. At short times after mitosis, the post-mitotic lag time and time inhomogeneity of the bursting kinetics can contribute to 50-75% of the peak Fano. At steady state, the only source of noise is bursting kinetics (see Figure 3B,C,G,H)”.

6. Lines 367-368 state “We hypothesize that this difference in initiation rates may contribute to the increased transcriptional noise elicited by the TATA box.” However, the mechanistic connection between higher initiation rates and increased noise is not clearly established. The authors' previous work (Pimmett et al., 2021) demonstrated that TATA boxes increase Pol II initiation rates and transcriptional levels, but simply observing that both initiation rates and noise are elevated doesn't establish causality. Is the argument that higher initiation rates inherently lead to more noise, or is this simply a correlation with overall transcriptional activity? If noise simply correlates with transcriptional activity, then the TATA box findings may be less informative than suggested.

As already stressed, in general, noise does not correlate with transcriptional activity. In order to mathematically prove this, we have added to the Methods analytical formulas for two state and three state models that relate steady state Fano and mean transcriptional activity to kinetic parameters. These formulas clearly show that Fano and the mean transcriptional activity are independent (Eqs.8-11). Furthermore, the same conclusion can be drawn from Figure S5, that plots the Fano vs the mean transcriptional activity for all our phenotypes. Noise depends on all the kinetic parameters in a way different from the mean activity.

However, our statement is correct because, for this specific pair of genotypes, the increase in noise is driven primarily by an increase in k_{ini} . Using the analytical formulas (detailed line 971) and the Supplementary Table 3 we were able to predict the Fano factor at steady state and the values are in good agreement with the observations.

Doing the arithmetic, Eq.8 (which gives the Fano factor for the three-state model) shows that the numerator $k_{1on}^2 k_{2off} + k_{2on}^2 k_{1off}$ and the denominator $k_{1on} k_{2on} + k_{1on} k_{2off} + k_{2on} k_{1off}$ change only slightly between genotypes, because variations in the parameters entering these expressions occur in opposite directions and largely cancel. Therefore, for these two genotypes, the variation in the Fano factor is driven primarily by k_{ini} . We added the full derivation of the analytical formulas and this numerical application as a supplementary text and a summary line 971.

Minor Comments

Line 123 - need "used" in "We the MS2/MCP mRNA labeling system and.."

thanks, corrected

Fig 5F - the WT twi should have no "x" on DPE, as the twi enhancer contains DPE, no?

Thanks, this is now corrected.

I didn't put everything, but there were other typos and errors in the manuscript and figures.

We did our best to correct these.

Second decision letter

MS ID#: dev.204953R1

MS TITLE: Mitotic reactivation and transcriptional bursting govern transcriptional noise in the early *Drosophila* embryo

AUTHORS: Mounia Lagha, Louise Maillard, Virginia L. Pimmett, Maria Douaihy, Pablo Garcia Idieder, Rachel Topno, Amelie Brun, Antonio Trullo and Ovidiu Radulescu

Dear Dr Lagha,

I am happy to tell you that your manuscript has been accepted for publication in *Development*, pending our standard publication integrity checks.

Reviewer 1

Advance summary and potential significance to field

We would like the authors for responding thoroughly to the questions posed in our original review. The rebuttal and proposed edits are satisfactory in addressing the points we had raised. In its revised format, the manuscript is of high quality and we recommend the article for publication.

Reviewer 3

Advance summary and potential significance to field

They have addressed most of my concerns, and I endorse the publication of this manuscript in *Development*.